# CAN LANGUAGE MODELS COMPOSE SKILLS DEMONSTRATED VIA IN-CONTEXT EXAMPLES?

## ABSTRACT

Composing basic skills from simple tasks to accomplish composite tasks is crucial for modern intelligent systems. We investigate the *in-context composition* ability of language models to perform composite tasks that combine basic skills demonstrated only in in-context examples. This is more challenging than the standard setting, where skills and their composition can be learned in training or from contextual information. We conduct systematic experiments on various representative open-source language models, utilizing linguistic and logical tasks designed to probe composition abilities. The results reveal that simple task examples can have a surprising *negative impact* on the performance, because the models generally struggle to recognize and assemble the skills correctly, even with Chain-of-Thought examples. Theoretical analysis further shows it is crucial to align examples with the corresponding steps in the composition. This inspires a method for the probing tasks, whose improved performance provides positive support for our insights.

## 1 INTRODUCTION

Recent advances in machine learning have yielded substantial progress, particularly with the rise of language models (e.g., (OpenAI, 2023; Anthropic, 2024; DeepSeek-AI, 2025)). These models exhibit strong in-context learning (ICL) capacity: they can adapt to novel tasks by leveraging a few examples provided at inference time without requiring parameter updates. Through ICL, language models can generalize across tasks by adapting to the given context. A critical aspect of this task generalization is the ability to integrate basic skills from simple tasks to perform more complex composite ones, which is essential given the exponential number of possible compositions that prevents learning each individually. Ideally, models should be able to compose skills demonstrated in-context to tackle new compositions. This leads to our central question: *Can language models do composition in-context?*

This study proposes to study the *in-context composition* ability of language models. Specifically, it examines whether models can solve queries for novel composite tasks combining several unknown simple tasks, when provided with some in-context examples from the simple tasks and some examples from the composite task. Unlike traditional scenarios where the skills and potentially some of their compositions are learned during training, the models in this study need to learn novel skills and compositions during inference, thus demanding strong compositional generalization.

We first perform systematic empirical studies on representative language models (Touvron et al., 2023a;b; Karamcheti et al., 2021; Grattafiori et al., 2024; Guo et al., 2025a), using linguistic and logical tasks from Xu et al. (2024b) designed for probing the composition abilities. The experiments show a surprising phenomenon: *simple task examples can hurt the performance on composite queries, rather than improve it*. See an illustration in Fig. 1. This is in sharp contrast to the expectation that these examples can help the model identify skills and compose them to solve the query. Our investigation of this negative impact finds that the models generally do not recognize the composition and do not align the simple task examples with the corresponding steps of the composition. Even when Chain-of-Thought (CoT) examples are used, they may mismatch the skills inferred from examples to the wrong steps in answering the composite query. Inspection into the inner attentions of the language models provides further evidence for our findings.

We further provide a theoretical analysis in a stylized setting that captures the essence of the in-context composition and focuses on understanding the key factor behind the observations. The analysis confirms that ignorance of the compositional structure can harm the performance, while *aligning*

*the examples to appropriate steps of the composition* can potentially improve it. This inspires a proof-of-concept algorithm for the probing tasks, Expanded Chain-of-Thought (ExpCoT), that views simple task examples as composite task examples with missing steps and expands them into the CoT format with missing steps marked by special symbols. Evaluations show that it can explicitly align the examples with the corresponding steps and thus improve the performance. The improvement verifies our insights and justifies the potential for helping future algorithm designs.

Our contributions are summarized as follows.

- We perform systematic experiments investigating the in-context composition ability of representative open-source language models and find that they typically exhibit limited such ability, due to difficulties in recognizing the composition and identifying proper skills from in-context examples.

- We provide a theoretical analysis, which explains the empirical observations and reveals that explicitly aligning in-context examples with the corresponding steps can help accomplishing the composite tasks.

- We propose a proof-of-concept method that significantly improves the in-context composition performance on the probing tasks, which provides positive support for our insights.

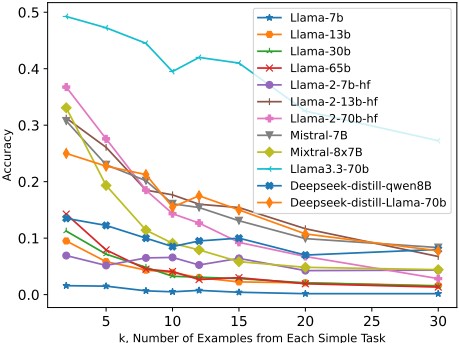

Figure 1: The negative impact of simple task examples on the opposition+swap task (see Table 1). The models need to answer a composite query when given $k$ examples from each simple task and $k_c = 5$ examples from the composite task. They show unexpected *decreasing* performance with more simple task examples ($k$).

## 2 RELATED WORK

**In-Context Learning and Chain-of-Thought.** Several studies investigate the behavior of in-context learning (ICL). Zhao et al. (2021); Lu et al. (2022); Min et al. (2022b); Wei et al. (2023) analyze the sensitivity of LLMs to in-context examples. Rubin et al. (2022); Liu et al. (2022); Hongjin et al. (2023); Wang et al. (2023a) propose methods to effective selection of in-context learning examples. Garg et al. (2022); Von Oswald et al. (2023); Akyürek et al. (2023); Mahankali et al. (2023); Zhang et al. (2023a); Shi et al. (2024) investigate with linear models, showing how transformers can represent gradient descent and conduct linear regression. Guo et al. (2024) provide analysis on how ICL works in non-linear functions. Chain-of-Thought (CoT) prompts LLMs to produce intermediate reasoning steps to solve multi-step reasoning questions Kojima et al. (2022); Wei et al. (2022). Few-Shot-CoT improves LLM's reasoning ability using demonstrations that are either manually constructed Khot et al. (2022); Zhou et al. (2023); Li et al. (2023); Wang et al. (2023b) or automatically selected Zhang et al. (2023b). Several theoretical work have been proposed to analyze the effecitiveness of CoT. Feng et al. (2023); Li et al. (2024) shows CoT allows for performing more serial computations, increasing the effective depth of a transformer. Joshi et al. (2025) presents a frameworks that allows for universal representability and computationally tractable CoT learning and Abedsoltan et al. (2025) analyzes the task generalization enabled by composition. Our theoretical analysis is partially inspired by Joshi et al. (2025); Abedsoltan et al. (2025) but considers a different setting with both simple/composite task examples, and also analyzes when the composition can fail.

**Compositional Task Learning.** Compositional reasoning of LLM is an active area in AI Huang & Chang (2022); Sinha et al. (2024). Kim & Linzen (2020); Levy et al. (2022) explore the compositional capabilities of LLMs in abstract reasoning tasks under ICL settings. An et al. (2023a;b) show LLMs are capable of learning abstract reasoning (e.g., grammar) to perform new tasks when finetuned or given in-context examples. Ye et al. (2023); Dziri et al. (2023); Thomm et al. (2024); Xu et al. (2024b) show that LLMs can handle simple sub-tasks, but often struggle with tasks composing multiple sub-tasks. Press et al. (2023) show that challenge in composition can be mitigated through CoT prompting. (Zhao et al., 2024) demonstrates that small-scale LLMs can learn and generalize compositional skills through finetuning on tasks involving skill combinations. (Song et al., 2025;

Yang et al., 2024b; Brinkmann et al., 2024; Guo et al., 2025b; Hong et al., 2024) provide mechanistic analyses on how LLMs tackle compositional reasoning. (Chen et al., 2024) also studied composing skills in-context, with the focus of unlocking the compositionality ability of the model. It provided carefully designed Skills-in-Context Prompting, which includes explanations of the basic skills along with examples, and also step-by-step explanations about how to compose them to solve the compositional query. Such a prompting allows the model to more actively utilize pre-existing internal skills from pretraining for compositional tasks. (He et al., 2025) studied the setting where the model can predict the required skills for the query and retrieve in-context examples from a given pool. It noted that such skill-based prompting can hurt small model performance on easy questions by introducing unnecessary information and resulting in overthinking, and then provided an adaptive method to address this issue. Our work conducts systematic experiments for in-context composition ability (whether the model can infer and compose skills demonstrated only via in-context examples) on the tasks from Xu et al. (2024b) and provides empirical/theoretical analysis on the reasons for success and failure. More discussions on related work are in Appendix A.

## 3   EMPIRICAL STUDY ON THE IN-CONTEXT COMPOSITION ABILITY OF LLMs

To examine the in-context compositional capabilities of language models, we conduct systematic experiments utilizing public large language models on linguistic and logical composite tasks.

**Models and Dataset.** We use 12 models: Llama (7B, 13B, 30B, and 65B) (Touvron et al., 2023a), Llama2 (7B, 13B, and 70B) (Touvron et al., 2023b), Mistral (7B and 8x7B) (Karamcheti et al., 2021), Llama3.3 (70B) (Grattafiori et al., 2024), Deepseek-distill (-qwen8B and -Llama2-70b) (Guo et al., 2025a). We adopt the test suite with nine composite tasks from Xu et al. (2024b). The simple tasks apply functional mappings to words represented by operators like $*$, while composite tasks combine two simple tasks, as illustrated in Table 1. Appendix B.1 includes the dataset details like the list of tasks, the numbers of queries tested, etc. Our code will be made public.

These tasks fit our purpose of investigating in-context composition. (1) They are clean compositions of linguistic and logical skills that allow in-depth investigation. (2) They are at appropriate levels of difficulty, while too simple composite tasks will be easily solved and too difficult ones will lead to consistently low performance, which will obscure interesting observations and prevent detailed examinations. (3) They construct synthetic test data with customized syntax operators to ensure novel tasks (see discussion in Xu et al. (2024b)). While the models have corresponding linguistic/logic abilities (e.g., they can answer queries like "what's the opposition of the word dry"), the operators have not been associated with the linguistic and logical tasks in pretraining, so the model needs to learn the simple tasks and their composition in-context at inference time. Furthermore, the models can learn the simple tasks via a few examples (see Figure 2 in Xu et al. (2024b)), which thus allows investigation of whether they can learn composition via examples.

|  | Simple Task 1 (opposition) | Simple Task 2 (swap) | Composite Task | In-Context Composition |
|---|---|---|---|---|
| Prompt | input: * Dry Lie | input: Sad Less # | input:* Eager Proud # | input: * Dry Lie
output: Wet Stand
input: Sad Less #
output: Less Sad
input: * Eager Proud #
output: Humble Listless
input: * Rich Humble # |
| Answer | output: Wet Stand | output: Less Sad | output: Humble Listless | output: Proud Poor |

Table 1: An example of In-context composition. Here the composition consists of two simple tasks: opposition (represented by operator $*$) and swap (represented by operator $\#$).

**Experimental setup.** Consider a composite task combining two simple tasks (task 1 and task 2). The test prompt given to models will consist of in-context examples (including $k_1$ examples from task 1, $k_2$ examples from task 2, and $k_c$ examples from the composite task), and a query from the composite task. The examples drawn from simple tasks demonstrate basic skills, while the composite examples illustrate how these skills can be integrated. This setup evaluates whether the model can solve new composite queries by utilizing examples of the individual skills and also their composition. We also perform additional experiments on the composition of more than 2 tasks in Appendix B.9 and on GPT4.1/Gemini2.5 in Appendix B.10, which provides further support for our analysis.

LLMs are known to be sensitive to the orders of in-context examples (e.g., Lu et al. (2022)). To avoid the influence on our investigation, we randomly shuffle the $k_1 + k_2 + k_c$ examples (see Appendix B.1.1 for an ablation study). Each prompt is evaluated with 4 different random shufflings, and we report the average accuracy across both test prompts and random seeds.

**Result summary.** We investigate the following questions: (**Q1**) Can in-context examples help composition? (**Q2**) What information from the in-context examples is utilized? (**Q3**) How are the in-context examples utilized? (**Q4**) Where are the models paying attention to? (**Q5**) Can Chain-of-Thought examples help? Our experiments provide the following findings. (**A1**) By increasing the number of simple or composite task examples, we observe that composite ones help while simple task examples unexpectedly hurt the performance. (**A2**) By examining the outputs, we find that the model may ignore the compositional structure: it may match the composite query to examples from any task and perform only the matched task. Then more examples from a simple task lead to higher chance of performing only that simple task on the query and thus worse performance. (**A3**) By an ablation study on different parts of the examples, we find that the models largely match the query to examples based on the operators rather than the semantic content. (**A4**) By visualizing the inner attention map, we illustrate that the models typically do not recognize the composition and pay roughly equal attention to the simple and composite examples. (**A5**) By adding intermediate step outcomes (CoT) in the composite task examples, we find that naïve CoT may not help, because the model may not align the examples to the corresponding steps in the composition. Below we present our empirical studies. Due to space limitations, some details/results are deferred to the appendix.

### 3.1 COMPOSITE TASK EXAMPLES HELP BUT SIMPLE TASK EXAMPLES HURT COMPOSITION

Ideally, the model can learn the basic skills demonstrated in the simple task examples, and learn how to compose the skills from the composite task examples. In this case, increasing the number of simple task examples or composite task examples should enhance the performance on composite queries. To evaluate the impact of in-context examples, we vary the number of examples provided to the model. Specifically, we set the number of examples for each simple task to be equal ($k_1 = k_2 := k$), and then evaluate performance across different $k \in \{2, 5, 8, 10, 12, 15, 20, 25, 30\}$ and $k_c \in \{0, 1, 2, 5, 10\}$.

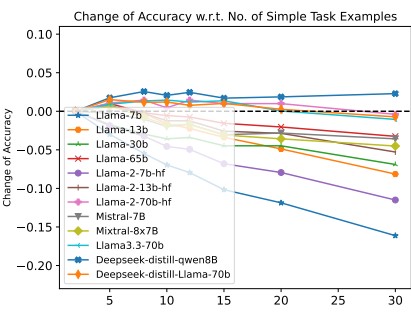
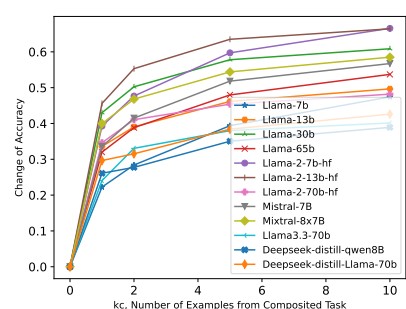

(a) Increasing the number of simple examples  (b) Increasing the number of composite examples

Figure 2: The effect of the in-context examples on in-context composition. The average **change** of the accuracy is reported, averaged over the tasks and $k_c$ (or $k$). More simple task examples surprisingly harm the composition performance, while more composite task examples help as expected.

**Results.** Fig. 2 shows the *change* of the accuracy with increasing $k$ or $k_c$, to highlight the trend (detailed accuracy results themselves are included in the appendix). It reveals an unexpected pattern: while performance on composite queries improves when we increase the number of composite task examples ($k_c$), it surprisingly declines as we add more simple task examples ($k$). This implies, more examples of simple tasks can be harmful rather than helpful for the composite task. This counterintuitive result highlights the sophistication of the in-context composition.

More precisely, Fig. 2(a) shows that the accuracy of the models (averaged over all tasks and $k_c$ values) typically decreases when the number of simple task examples $k$ increases. For instance, the average accuracy of the model Llama-13B drops by 7.5% when $k$ increases from 2 to 30. This trend is consistent across most settings (different models/tasks/numbers of examples); see additional details including breakdowns by individual task and $k_c$ in Appendix B.2. Deepseek-distill-qwen8B appears

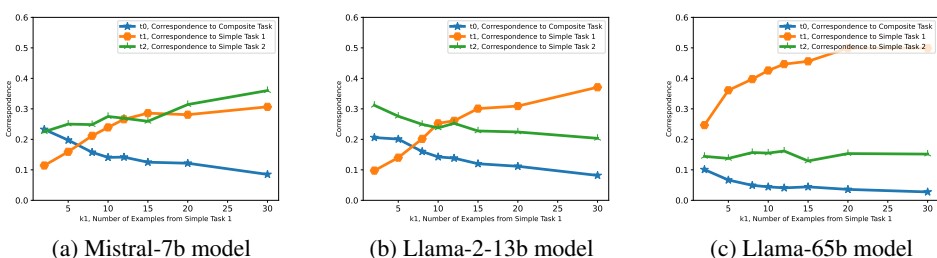

| (a) Mistral-7b model | (b) Llama-2-13b model | (c) Llama-65b model |

Figure 3: The output distribution on opposition+swap for different numbers of task 1 examples ($k_1$). The correspondence to task 1 increases, while those to task 2 and the composite task decrease.

less affected, but a detailed look finds that this is because its accuracy saturates on some easier tasks for larger $k_c$; it can be negatively affected on some other tasks. In contrast, Fig. 2(b) shows that more composite examples clearly improve the performance, which is also consistent in most settings.

These results suggest that the model fails to recognize the composition and cannot utilize simple task examples, treating them as interfering noise rather than useful signals. We further test on queries from simple tasks and find that more composite task examples also decrease the accuracy on these queries (see results in Appendix B.2.1), supporting our hypothesis that the models do not recognize the composition. This motivates our subsequent investigations into the mechanisms underlying how models extract and process information from in-context examples.

### 3.2 THE MODELS MAY IGNORE THE COMPOSITIONAL STRUCTURE

The above observations suggest the model can utilize the composite examples properly but not the simple task examples. We would like to check what information from the examples is utilized and how. A detailed look into the outputs shows that the error is often by performing only one simple task on the composite queries. For example, for an opposition+swap task query, the model only performs the opposition task and generates the output. To quantify this, we measure the outputs' correspondence to performing different tasks: the correspondence to the composite task (denoted as $t_0$) is the accuracy w.r.t. the answer by performing the composite task, the correspondence to simple task 1 (denoted as $t_1$) is the accuracy w.r.t. the answers by performing task 1, and similarly for task 2 (denoted as $t_2$). We then measure these correspondences for different task 1 example numbers $k_1$.

**Results.** Fig. 3 shows the results for three models on the opposition+swap task; see additional results in Appendix B.3. With more task 1 examples, the correspondence w.r.t. the task 1 increases, while those for task 2 and the composite task decrease. This suggests that the models may match the composite query to in-context examples from any task, and the matching probability is proportional to the number of examples from each task. When the number of task 1 examples $k_1$ increases, the model becomes more likely to match the query to task 1 examples, explaining the increased task 1 correspondence. For further verification, we conduct experiments with increasing $k_c$; see Appendix B.3. With more composite examples, the correspondence to the composite task increases while the others decrease. Furthermore, we repeat the experiments without task 2 examples ($k_2 = 0$) and observe the same trend. These observations provide further support for our above analysis.

### 3.3 THE MODELS LARGELY UTILIZE THE OPERATORS RATHER THAN THE CONTENT

Here we investigate how the models explore the examples by controlled experiments that alternate different parts of the examples. As shown in Table 1, the examples contain two parts: (1) the operators like ⋆ and # that denote the tasks to be accomplished, and (2) the content, i.e., the input and output text demonstrating the operation performed. We first introduce an irrelevant task, and then replace the content or operator in the composite task examples with that in the irrelevant task examples. This gives two ablation settings (illustrated in Table 2): (1) Irrelevant content that replaces the content; (2) Irrelevant operator that replaces the operators. Evaluations in these two settings can reveal the influence of the content/operator on utilizing the examples.

|  | Original | Irrelevant Task | Irrelevant Content | Irrelevant Operator |
|---|---|---|---|---|
| Prompt | input: * Dry Lie | | input: * Dry Lie | input: * Dry Lie |
| | output: Wet Stand | | output: Wet Stand | output: Wet Stand |
| | input: Sad Less # | | input: Sad Less # | input: Sad Less # |
| | output: Less Sad | | output: Less Sad | output: Less Sad |
| | input: * Eager Proud # | input: ( Accept Low ) | input: * Accept Low # | input: ( Eager Proud ) |
| | output: Humble Listless | output: ACCEPT LOW | output: ACCEPT LOW | output: Humble Listless |
| | input: * Rich Humble # | input: ( Rich Humble ) | input: * Rich Humble # | input: ( Rich Humble) |
| Answer | output: Proud Poor | output: RICH HUMBLE | output: RICH HUMBLE | output: Proud Poor |

Table 2: Illustrations of the two ablation settings for opposition+swap. The irrelevant content/operator setting replaces the original content/operator with that from the irrelevant task capitalization.

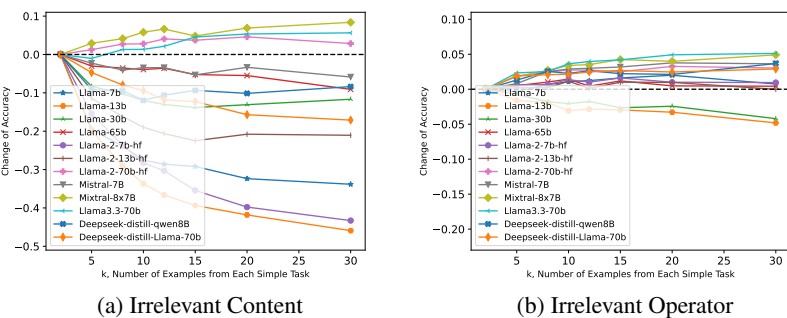

(a) Irrelevant Content  (b) Irrelevant Operator

Figure 4: Results for ablating the content/operators in the composite task examples. Increasing $k$ still affects the performance after ablating the content, but has little impact after ablating the operators.

**Results.** Fig. 4 shows that in the irrelevant operator setting, increasing $k$ has little impact after ablating the operators. This suggests that the model almost ignores the simple examples after using a different operator, i.e., the utilization of the examples is largely based on the operators. While in the irrelevant content setting, increasing $k$ still affects the performance after ablating the content. Note that the performance trends of some models are changed after ablating the content, so the content still plays a role in utilizing the examples, but the role is less clear and significant. More detailed results in Appendix B.4 provide further support.

### 3.4 INNER ATTENTIONS MAY NOT DISTINGUISH THE SIMPLE AND COMPOSITE TASKS

**Similarities between attentions for simple and composite queries.** We compare the inner attentions of the model facing simple or composite task queries. Following Hong et al. (2024), we consider oppopistion+swap and choose 100 test prompts with simple task queries and 100 test prompts with composite task queries. Then we choose a layer, extract the attention output for each query. Finally, we compute the pairwise cosine similarities between the attentions, giving a 200 by 200 similarity matrix. To generate the prompts, we consider a fixed context with simple/composite task examples ($k = 10$ and $k_c = 5$), and then add different queries (100 simple and 100 composite ones).



Figure 5: Similarities of the attentions between 100 composite queries (first 100 rows/columns) and 100 simple queries (last 100 rows/columns). Attentions are from Layer 15, 17, and 19 of Mistral-7B.

Fig. 5 shows that the inner attentions have high similarities between the simple task queries and composite task queries. In fact, the similarities between the two groups are at the same level of those within each group. This suggests the model cannot distinguish between simple and composite tasks. More quantitative results in Appendix B.5.1 provides further evidence, e.g., the entropy distributions of the attentions for simple/composite queries do not have a significant difference.

## 3.5 CHAIN-OF-THOUGHT EXAMPLES MAY NOT HELP

Chain-of-Thought (CoT) is popular for multi-hop reasoning and fits our setting, i.e., splitting the composite task examples into steps, each performing only one simple task. Consider an example `* Rich Humble # -> Proud Poor` where `*` denotes opposition and `#` denotes swap. Then the corresponding CoT example is `* Rich Humble # -> Poor Proud # -> Proud Poor`, which now includes the intermediate output `Poor Proud #`. We replace all the composite examples with their CoT version and redo the experiment in Section 3.1, to examine if CoT can help in-context composition.

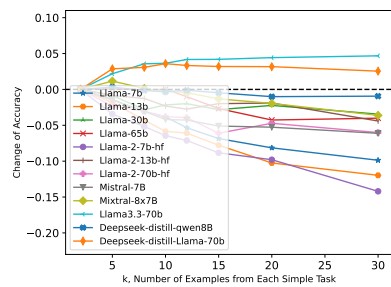

Figure 6: CoT cannot help mitigate the negative impact from adding more simple task examples.

**Results.** Fig. 6 shows larger $k$'s still lead to worse performance, i.e., CoT does not help mitigate the negative impact of simple task examples, and thus does not help utilize the examples. Furthermore, for a fixed $k$, the performance may not improve over that without CoT (shown in Table 3 below). We check the details of the output, and find the model typically performs a Chain-of-Thought and generates two step outputs, but in the intermediate step it may match with wrong examples and perform wrong operations as before. Consider an opposition+swap query `* Grow Respect #`. The expected output is `* Grow Respect # -> Shrink Disrespect # -> Disrespect Shrink`. However, the model outputs `* Grow Respect # -> Shrink Disrespect # -> Respect Grow`. In particular, in the second step, the model incorrectly performs both tasks opposition+swap, while it should perform only swap. So applying CoT naïvely may not help recognize the composition and align basic skills properly.

## 4 THEORETICAL ANALYSIS

The empirical study reveals the crucial role of recognizing the composition and matching the basic skills with corresponding steps in the composition. This section further provides some theoretical analysis on whether it is feasible for the model to accomplish the composite task given the in-context examples. Since large language models have universal expressive power to represent various algorithms (e.g., Giannou et al. (2023); Malach (2024)), we reduce the research question to the existence of learning rules that can achieve small errors on the composite task queries given simple and composite task examples.

**Theoretical setup.** A sequence-to-sequence task on a finite vocabulary of tokens $\Sigma$ is associated with an input distribution $\mathcal{D}$ over the input $\mathcal{X} \subseteq \Sigma^*$, and a target function $f : \Sigma^* \to \Sigma^*$ where $f \in \mathcal{H}$ for some model class $\mathcal{H}$. A composite task with the target function $f \in \mathcal{H}^T$ can consist of $T$ steps $f_1, f_2, \ldots, f_T \in \mathcal{H}$, such that $f(x) = f_T \circ \cdots \circ f_2 \circ f_1(x)$. For simplicity, assume $\mathcal{H}$ is finite, and it includes the identity mapping so that $\mathcal{H} \subseteq \mathcal{H}^T$.

**Learning on composite task examples.** We first consider the case when $k_c$ composite task examples $\mathcal{S}_0 = \{(x_i, y_i) : i \in [k_c]\}$ are given, where $x_i$ are i.i.d. from $\mathcal{D}$ and $y_i = f(x_i)$ for some $f \in \mathcal{H}^T$. Since $\mathcal{H}^T$ is finite, applying a standard generalization argument on $\mathcal{H}^T$ can show that a large enough $k_c$ allows accomplishing the task (proof in Appendix C).

**Proposition 1.** *There exists a learning rule $\mathcal{M} : (\mathcal{X} \times \Sigma^*)^* \to \Sigma^{\mathcal{X}}$ such that for any distribution $\mathcal{D}$ over $\mathcal{X}$ and any $f \in \mathcal{H}^T$, for every $0 < \delta < 1$, we have with probability at least $1 - \delta$ over $\mathcal{S}_0$, $\Pr_{x \sim \mathcal{D}}[\mathcal{M}(\mathcal{S}_0)(x) \neq f(x)] \leq \frac{1}{k_c} \left(T \ln |\mathcal{H}| + \ln\left(\frac{1}{\delta}\right)\right)$.*

This ignores the compositional structure of the task, so the error increases fast with the number $T$ of steps in the composition: it increases linearly with $T$. Furthermore, to build general intelligent systems for various composite tasks (in exponential number $|\mathcal{H}|^T$), it is infeasible to learn from scratch on each individually. We thus aim at the compositional ability: break all composite tasks into simple tasks (i.e., small $|\mathcal{H}|$), learn basic skills on simple task examples, and learn how to compose for a target composite task with a few composite task examples. We next turn to such scenarios.

**Learning on examples from simple and composite tasks.** Suppose $\mathcal{S}_t$ is a set of examples from the $t$-th task $(\mathcal{D}_t, f_t)(0 \le t \le T)$. Suppose a method $\mathcal{M}(\mathcal{S}_0; \mathcal{S}_1, \ldots, \mathcal{S}_T)$ can focus on the 0-th task with the help of examples from the other tasks. Formally, we say $\mathcal{M}$ is *focusing* if its expected error on the 0-th task is no worse than that on any other task, i.e., for any $j \in [T]$,

$$\mathcal{L}_0(\mathcal{M}; (\mathcal{D}_t, f_t)_{t=0}^T) \le \mathcal{L}_j(\mathcal{M}; (\mathcal{D}_t, f_t)_{t=0}^T) \tag{1}$$

where $\mathcal{L}_j(\mathcal{M}; (\mathcal{D}_t, f_t)_{t=0}^T) := \mathbb{E}_{\mathcal{S}_t \sim (\mathcal{D}_t, f_t), 0 \le t \le T} \Pr_{x \sim \mathcal{D}_j}[\mathcal{M}(\mathcal{S}_0; \mathcal{S}_1, \ldots, \mathcal{S}_T)(x) \ne f_j(x)]$ is the expected error of $\mathcal{M}$ on the $j$-th task. When $f_0$ is a composite task composing $f_1, \ldots, f_T$, ideally, the method should use simple task examples to help learn each step in the composition. However, empirical studies show that the method may not distinguish between the two types of examples. Formally, we say that $\mathcal{M}$ does *not distinguish examples from different tasks*, if it is symmetric w.r.t. the data sets $\mathcal{S}_t$'s, i.e., for any permutation $\sigma$ on $\{0, 1, \ldots, T\}$, the distribution of $\mathcal{M}(\mathcal{S}_{\sigma(0)}; \mathcal{S}_{\sigma(1)}, \ldots, \mathcal{S}_{\sigma(T)})$ is the same as that of $\mathcal{M}(\mathcal{S}_0; \mathcal{S}_1, \ldots, \mathcal{S}_T)$. We derive a lower bound for its error:

**Proposition 2.** *Suppose there exist $g_1, \ldots, g_T \in \mathcal{H}$ with pairwise difference at least $\Delta$ for some $\mathcal{D}$, i.e., $\min_{i \ne j} \Pr_{x \sim \mathcal{D}}[g_i(x) \ne g_j(x)] \ge \Delta$. For any $\mathcal{M}$ that is focusing but does not distinguish between examples from different tasks, there exist $f_1, \ldots, f_T \in \mathcal{H}$, $f_0 = f_T \circ \cdots f_2 \circ f_1$, and $\mathcal{D}_t(0 \le t \le T)$'s, such that $\mathbb{E}_{\{\mathcal{S}_t\}} \Pr_{x \sim \mathcal{D}_0}[\mathcal{M}(\mathcal{S}_0; \mathcal{S}_1, \ldots, \mathcal{S}_T)(x) \ne f_0(x)] = \Omega(\Delta)$.*

The result shows that when the model class is reasonably rich, there are always cases where the method fails (the error can be as large as the diameter of the model class). Intuitively, such a method may mistakenly confuse simple task examples as data for the whole composition, in which case the examples act as harmful noise as seen in our experiments. Similar observations for naïve CoT, i.e., composite task examples consisting of the intermediate outputs in the composition. So it is crucial to present the examples in a way that can let the method distinguish between simple and composite task examples and align the simple task examples with the proper step in the composition.

Now suppose the method knows that $\mathcal{S}_t(t \in [T])$ are examples for the $t$-th step of the composition. Furthermore, suppose $\mathcal{S}_0$ are CoT examples from the composite task, i.e., each example is in the form $(z^1, z^2, \ldots, z^{T+1})$ where $z^1$ is the input $x$ and $z^{t+1} = f_t(z^t)$ are the intermediate output for $t \in [T]$. We show that such examples have the potential to help the composition.

**Theorem 1.** *Suppose we are given $k_t$ examples $\mathcal{S}_t$ from $(\mathcal{D}_t, f_t)$ for $f_t \in \mathcal{H}(t \in [T])$ and $k_c$ examples $\mathcal{S}_0$ from $(\mathcal{D}_0, f_0)$ with $f_0 = f_T \circ \ldots \circ f_2 \circ f_1$. Suppose $\mathcal{H}$ is distinguishable: for some $\epsilon_0 > 0$, for any $f \ne g \in \mathcal{H}$ and $\mathcal{D}_t(0 \le t \le T)$, $\Pr_{x \sim \mathcal{D}_t}[f(x) \ne g(x)] > \epsilon_0$. There exists a learning rule $\mathcal{M} : ((\mathcal{X} \times \Sigma^*)^*)^{T+1} \to \Sigma^{\mathcal{X}}$ such that for every $0 < \delta < 1$, if $\max(k_c, k_t) \ge \frac{1}{\epsilon_0}\left(\ln |\mathcal{H}| + \ln \frac{T}{\delta}\right), \forall t \in [T]$, then with probability at least $1 - \delta$ over $\{\mathcal{S}_t\}_{t=0}^T$, we have $\mathcal{M}(\mathcal{S}_0; \mathcal{S}_1, \ldots, \mathcal{S}_T) = f_0$.*

The theorem shows that by exploiting the compositional structure, the sample size needed is logarithmic in $T$ (compared to linear in Proposition 1). Furthermore, the $k_t$ examples from simple task $t$ is now useful for the identification of the $t$-th step. This inspires a new method below.

### 4.1 Verification of the insights: the expanded Chain-of-Thought method

Inspired by insights from our analysis, this section introduces a novel variant of CoT for improving the in-context composition. The main idea is to view the simple task examples as composite task examples with missing steps and expand them into the CoT format with missing steps marked by special symbols. This will explicitly align the examples for better utilization.

For description, recall the composite task consists of $T$ steps $f_1, f_2, \ldots, f_T$. A CoT example on input $x$ is $(z^1, z^2, \ldots, z^{T+1})$ where $z^1 = x$ and $z^{t+1} = f_t(z^t)$ for $t \in [T]$. We also have examples $(x^t, y^t)$ from the simple task $t$ where $y^t = f_t(x^t)$. Our method views $(x^t, y^t)$ as a composite task example $(z^1 = ???, \ldots, z^{t-1} = ???, z^t = x^t, z^{t+1} = y^t, z^{t+2} = ???, \ldots, z^{T+1} = ???)$ where ??? denotes missing entries. Algorithm 1 formally describes the method. It goes over all examples and adds the strings of steps to each example. For illustration, consider the composite task opposition+swap. A CoT example `* Rich Humble # -> Poor Proud # -> Proud Poor` can be viewed as ( `* Rich Humble #` , `Poor Proud #` , `Proud Poor` ), which is converted by our method to ( `Step1: * Rich Humble #` , `Step2: Poor Proud #` , `Step3: Proud Poor` ). Similarly, an example from the opposition task `* Dry Lie -> Wet Stand` is converted to ( `Step1: * Dry Lie` , `Step2: Wet Stand` , `Step3: ???` ). An example from the swap task `Sad Less # -> Less Sad` is converted to ( `Step1: ???` , `Step2: Sad Less #` , `Step3: Less Sad` ).

**Algorithm 1** Expanded Chain-of-Thought (EXPCOT)

**INPUT:** Chain-of-Thought examples $\mathcal{S}_0 = \{(z_i^1, z_i^2, \ldots, z_i^{T+1}) : i \in [k_c]\}$ from the composite task of $T$ steps, and examples $\mathcal{S}_t = \{(x_i^t, y_i^t) : j \in [k_t]\}$ from simple task $t \in [T]$
1: **for** $i \in [k_c], t \in [T+1]$ **do**
2:     $z_i^t \leftarrow$ `Step` $+\text{STR}(t)+$ `:` $+z_i^t$                          ▷ STR converts an integer into a string
3: **for** $t \in [T], i \in [k_t]$ **do**
4:     Replace $(x_i^t, y_i^t)$ with $(v_i^{t,1}, \ldots, v_i^{t,T+1})$, where $v_i^{t,j} \leftarrow$ `Step` $+\text{STR}(j)+$ `:` ??? for $j \notin \{t, t+1\}$, $v_i^{t,t} \leftarrow$ `Step` $+\text{STR}(t)+$ `:` $+x_i^t$, and $v_i^{t,t+1} \leftarrow$ `Step` $+\text{STR}(t+1)+$ `:` $+y_i^t$
**OUTPUT:** The updated data $\mathcal{S}_c, \mathcal{S}_t$ for $t \in [T]$

|  | L-7B | L-13B | L-30B | L-65B | L2-7B | L22-13B | L2-70B | M-7B | M-8x7B | L3-70b | D-8B | D-70B |
|---|---|---|---|---|---|---|---|---|---|---|---|---|
| Vanilla | 32.6 | 56.2 | 67.6 | 63.4 | **49.6** | 68.7 | 80.8 | 66.1 | 71.2 | 77.2 | 58.2 | 71.3 |
| CoT | 42.2 | 51.2 | 72.7 | 64.0 | 45.9 | 65.7 | 77.6 | 64.9 | 77.6 | **92.2** | 60.7 | 85.9 |
| ExpCoT | **47.5** | **58.1** | **77.4** | **75.7** | 47.9 | **70.4** | **87.2** | **74.3** | **87.5** | 91.3 | **75.1** | **88.7** |

Table 3: The accuracy (%) averaged over tasks ($k = 30, k_c = 2$). L: Llama, L2/3: Llama2/3.3, M: Mistral, D: Deepseek. Best results are **boldfaced**.

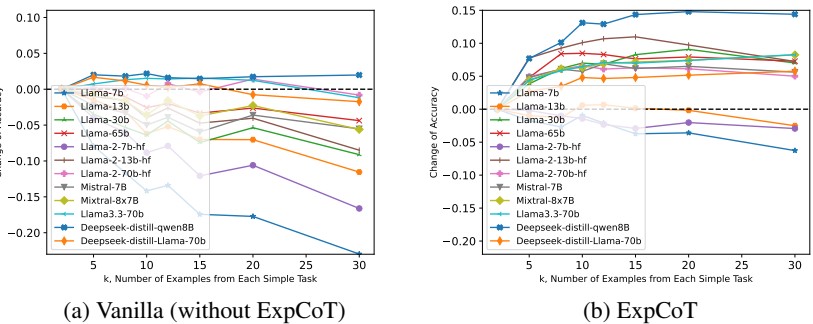

(a) Vanilla (without ExpCoT)                     (b) ExpCoT

Figure 7: The impact of more simple task examples for without or with ExpCoT ($k_c = 2$).

**Evaluation.** We first use Algorithm 1 on the examples and then redo the experiment in Section 3.1. Table 3 compares the average accuracy of without CoT, naïve CoT, and our ExpCoT, and shows that ExpCoT leads to significant improvement. We also compare the impact of simple task examples in the two cases of without and with ExpCoT in Fig. 7. With ExpCoT, the models can now utilize simple task examples better, except some small models likely because they still cannot identify the skills from the simple task examples due to the limited capacity. Failure analysis in Appendix B.8.1 shows that simple task confusion becomes rare since ExpCoT helps recognize the composition. These results demonstrate ExpCoT can improve the in-context composition.

## 5 CONCLUSIONS AND LIMITATIONS

This work studied the in-context composition ability of language models. Empirical studies of representative models on linguistic and logic tasks showed they in general have limited such ability due to the failure to recognize the composition and identify skills for the steps of the composition. Theoretical analysis showed that it is crucial to align skills from examples with the steps.

Note that typical text data may already have annotations for the basic skills, e.g., "by the Pythagorean Theorem", which can act as the annotation "Step1" in our ExpCoT method. Our studies suggest that such annotations can be crucial for the success of composition, and adding more such annotations can improve the performance. While annotation can be expensive for web-scale datasets if done via human supervision, one alternative way is to use LLMs to do the annotations and use the annotated data for self-boosting. Furthermore, it suggests synthesizing data with annotations to help the model learn to compose. These are an interesting research directions, which we will leave for future work. Also note that due to resource limitations, our empirical studies do not include the most powerful models like GPT-5, nor consider complex tasks like those targeted by AI assistants. The results from this work hopefully pave the road for investigations in more sophisticated models and tasks.

## ETHICS STATEMENT

Our work aims to improve the theoretical understanding of compositional tasks in in-context learning. Our paper is mostly academic in nature and we foresee no immediate negative ethical impact. We discover an unexpected phenomenon of LLM's compositional ability and provide detailed analyses of it, which may have a positive impact on the AI community. We hope our work will inspire effective algorithm design and promote a better understanding of the compositional ability of LLMs.

## REPRODUCIBILITY STATEMENT

For theoretical results in the Section 4, a complete proof is provided in the Appendix C. For experiments in the Section 3, complete details and experimental results are provided in the Appendix B. The source code with explanations and comments is provided in supplementary materials.

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

CONTENTS

# Appendix

## A  MORE DISCUSSION ON RELATED WORK

### A.1  LARGE LANGUAGE MODELS

LLMs are often Transformer-based (Vaswani et al., 2017) equipped with tmassive parameter sizes and extensive pretraining data OpenAI (2023); Anthropic (2024); DeepSeek-AI (2025); Yang et al. (2024a); Grattafiori et al. (2024). The training pipeline of LLM often involce pretraining and post-training. LLMs commonly adopt auto-regressive pretraining strategies (Radford et al., 2018; 2019; Brown et al., 2020). Significant research has focused on post-training methods to adapt LLMs for various tasks, such as multitask finetuning (Sanh et al., 2022; Wang et al., 2023c; Xu et al., 2024c), instruction tuning Chung et al. (2022); Mishra et al. (2022); Wang et al. (2022), in-context learning (Min et al., 2022b; Dong et al., 2022; Yao et al., 2023), and reinforcement learning from human feedback (RLHF) (Ouyang et al., 2022; Rafailov et al., 2023; Shao et al., 2024). As LLMs continue to scale in size, numerous studies have focused on improving their deployment efficiency, including memory management Xiao et al. (2024); DeepSeek-AI (2024); Dao et al. (2022); Dao (2024) and inference acceleration (Liang et al., 2024; Gu et al., 2024; Xu et al., 2024a; 2025).

### A.2  IN-CONTEXT LEARNING AND CHAIN-OF-THOUGHT

**In-context learning.** LLM exhibits a remarkable ability for in-context learning (ICL) (Brown et al., 2020; OpenAI, 2023; Team et al., 2023; Anthropic, 2024), which allows pretrained LLMs to solve specific tasks by conditioning on a few prepended in-context examples, without requiring any updates to the model parameters. Several empirical studies investigate the behavior of ICLs. Zhao et al. (2021); Lu et al. (2022) formulate the problems and analyze the sensitivity of LLMs to in-context examples sequences. Min et al. (2022b); Wei et al. (2023) investigate on how LLMs performance change react to the ground-truth label and text demonstrations within in-context examples. Rubin et al. (2022); Liu et al. (2022); Hongjin et al. (2023); Wang et al. (2023a) propose methods to effective selection of in-context learning examples. Chen et al. (2022); Min et al. (2022a) use meta training with an explicit in-context learning object to enhance performance. Theoretically, Xie et al. (2022) provide a bayesian framework to explain the working mechanism of in-context learning. Garg et al. (2022); Von Oswald et al. (2023); Akyürek et al. (2023); Mahankali et al. (2023); Zhang et al. (2023a); Shi et al. (2024) investigate with linear models, showing how transformers can represent gradient descent and conduct linear regression. Guo et al. (2024) provide analysis on how ICL works in non-linear functions. Based on these works, we present an analysis demonstrating how LLMs can exhibit compositional capabilities in ICL tasks.

**Chain-of-Thought reasoning.** Chain of thought (CoT) is widely used to solve multi-step reasoning questions Kojima et al. (2022); Wei et al. (2022). CoT generates an intermediate reasoning process in language before outputting the final answer. Typical CoT methods prompt LLMs to produce these intermediate steps either in zero-shot or few-shot settings. Zero-shot-CoT adds instructions such as "Let's think step by step" in prompts Kojima et al. (2022) while few-shot-CoT provides several examples with step-by-step reasoning as in in-context demonstrations Brown et al. (2020); Wei et al. (2022). Typical few-shot-CoT improves LLM's reasoning ability with manually designed demonstrations Khot et al. (2022); Zhou et al. (2023); Li et al. (2023); Wang et al. (2023b). Another line of research focuses on automatically selecting demonstrations, eliminating the need for manual construction Zhang et al. (2023b). Several theoretical work have been proposed to analyze the effeicitiveness of CoT. Liu et al. (2023) studies the expressiveness of shallow transformers, Feng et al. (2023); Li et al. (2024) further shows CoT allows for performing more serial computations that a vanilla transformer without CoT, increasing the effective depth of a transformer. Joshi et al. (2025) present a uniform framework that allows for universal representability and computationally tractable chain-of-thought learning. Abedsoltan et al. (2025) analyzes the task generalization enabled by composition. Our theoretical analysis is partially inspired by Joshi et al. (2025); Abedsoltan et al. (2025) but considers a different setting with both simple/composite task examples, and also analyzes when the composition can fail.

### A.3 COMPOSITIONAL TASK LEARNING

Solving complex tasks and reasoning of LLM is an active area of LLM research field Huang & Chang (2022); Sinha et al. (2024). There is a line of empirical works explored the compositional capabilities of LLMs in abstract reasoning tasks under ICL settings (Kim & Linzen, 2020; Levy et al., 2022). An et al. (2023a;b) show LLMs are capable of learning abstract reasoning (e.g., grammar) to perform new tasks when finetuned or appropriate in-context examples. Ye et al. (2023); Dziri et al. (2023); Thomm et al. (2024); Xu et al. (2024b) show that LLMs can handle simple sub-tasks, but often struggle with tasks that require composing multiple sub-tasks. Press et al. (2023) shows that such challenges can be mitigated through the use of chain-of-thought prompting. Berglund et al. (2024) reveals LLMs trained on relations like "A is B" fail to learn inverse relations "B is A". (Zhao et al., 2024) demonstrates that small-scale LLMs can learn and generalize compositional skills through fine-tuning on tasks involving skill combinations. (Song et al., 2025; Yang et al., 2024b; Brinkmann et al., 2024; Guo et al., 2025b; Hong et al., 2024) provide mechanistic analyses on how LLMs tackle compositional reasoning tasks. (Chen et al., 2024) also studied composing skills in-context, with the focus of unlocking the compositionality ability of the model. It provided carefully designed Skills-in-Context Prompting, which includes explanations of the basic skills along with examples, and also step-by-step explanations about how to compose them to solve the compositional query. Such a prompting allows the model to more actively utilize pre-existing internal skills from pretraining for compositional tasks. Our work focuses on investigating the models' ability in a prior unknown composition with a prior unknown skills, and thus considers the setting where skills are only demonstrated via in-context examples (without in-context explanations of the skills). (He et al., 2025) studied the setting where the model can predict the required skills for the query and retrieve in-context examples from a given pool. It noted that such skill-based prompting can hurt small model performance on easy questions by introducing unnecessary information and resulting in overthinking, and then provided an adaptive method to address this issue. Our work has a different focus on the in-context composition ability, i.e., whether the model can learn and compose the skills from the given in-context examples, so conducts experiments on the tasks from Xu et al. (2024b) and finds that LLMs fail on composite tasks when logical steps are intermixed. Our work further provides empirical and theoretical analysis on why the composition can succeed or fail and introduces an improved method.

## B EXPERIMENTAL DETAILS AND MORE RESULTS

### B.1 DETAILS OF THE DATASET AND SETUP

We use the dataset from Xu et al. (2024b), and construct 9 composite tasks for our experiments. The details can be found in Xu et al. (2024b), while here we provide some illustrations for convenience.

The composite tasks are compositions using eight simple tasks listed in Table 4. We use these simple tasks to construct the following composite tasks: opposition+swap (named 'oppopair swap' in the code), opposition+pastTense ('oppoverb'), pastTense+swap ('verbpair swap'), capitalization+swap ('upperswap'), swap+capitalization ('swapupper'), capitalization+twoSum ('upper twosum'), pastTense+plusOne ('verbsingle plusone'), pastTense+capitalization ('verbsingle upper'), plusOne+capitalization ('plusone upper').

**Experimental Setup.** For each composite task, the test prompts are generated using the code from the dataset. Four random seeds are used; for each random seed, $n$ test prompts are generated and the in-context examples in each test prompt are randomly shuffled. The number of test prompts $n$ is set to 100 for most composite tasks, except for two composite tasks with a small amount of data: $n$ is set to the maximum number 78 for opposition+pastTense, and $n$ is set to the maximum number 84 for pastTense+plusOne and pastTense+capitalization. Four NVIDIA H800 GPUs are used for the experiments.

### B.1.1 SHUFFLING V.S. NO SHUFFLING: THE EFFECT OF THE ORDER OF IN-CONTEXT EXAMPLES

The order of in-context examples is known to affect the performance of in-context learning. Here we perform an experiment comparing the results in four settings. (1) Shuffling: all the examples (simple and composite task examples) are randomly shuffled, and average accuracy over 4 random

| Tasks | Task | Input | Output |
|---|---|---|---|
| **Words** | (A) Capitalization | apple | APPLE |
| | (B) Swap | bell ford | ford bell |
| | (C) Two Sum | twenty @ eleven | thirty-one |
| | (D) Past Tense | pay | paid |
| | (E) Opposite | Above | Below |
| **Numerical** | (F) Plus One | 435 | 436 |
| | (G) Modular | 15 @ 6 | 3 |
| | (H) Two Sum Plus One | 12 # 5 | 18 |

Table 4: The collection of simple logical tasks. This table is adopted from Xu et al. (2024b).

seeds is reported. (2) Composition Last: the context consists of simple task 1 examples, followed by simple task 2 examples, and lastly the composite task examples. (3) Composition Middle: the context consists of simple task 1 examples, followed by the composite task examples, and lastly simple task 2 examples. (4) Composition First: the context consists of the composite task examples, followed by simple task 1 examples, and lastly simple task 2 examples.

Fig. 8 shows the results of two models Llama-2-7B and Mistral-7B on the opposition+swap task. The accuracies for the 4 settings are drastically different. This shows that the order of the examples indeed has a strong influence on the result. Such an influence can blur our investigation. Therefore, we randomly shuffle the examples to remove such an influence.

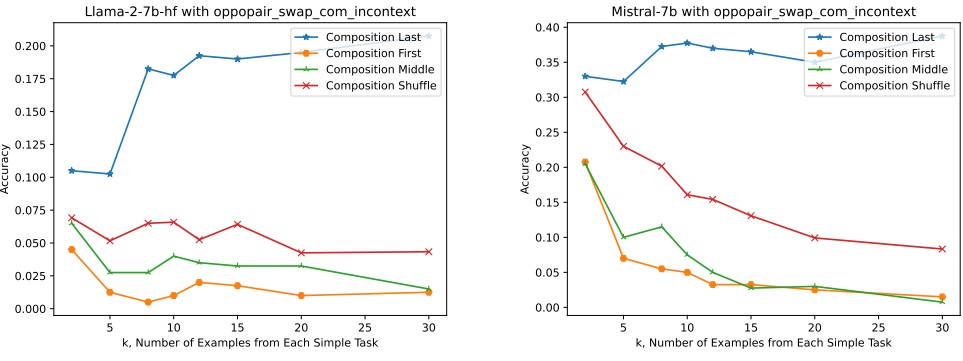

Figure 8: The effects of shuffling v.s. no shuffling.

## B.2 DETAILED RESULTS FOR THE EFFECT OF IN-CONTEXT EXAMPLES

In this section, we present detailed results from our experiments on the effect of in-context examples.

**In-context simple task examples.** Fig. 9 shows the effect of in-context simple task examples for each $k_c$ and model (the reported accuracy is averaged over tasks). More precisely, we draw a subfigure for each model and draw a curve for each $k_c$; the $x$-axis is the number $k$ of examples from each simple task, and $y$-axis is the accuracy averaged over all the composite tasks.

Fig. 10 shows the effect of in-context simple task examples for each task and model (the reported accuracy is averaged over $k_c$). More precisely, we draw a subfigure for each model and draw a curve for each task; the $x$-axis is the number $k$ of examples from each simple task, and $y$-axis is the accuracy averaged over all the $k_c$ values.

From the detailed results, we can see that, the larger models like Llama-2-70B and Mixtral-8x7B achieve quite high accuracies on many tasks when $k_c$ is large. The high accuracy do not change much for different $k$ and thus the negative impact of more examples from simple tasks is not significant. However, on harder tasks like opposition+swap, the negative impact is again substantial.

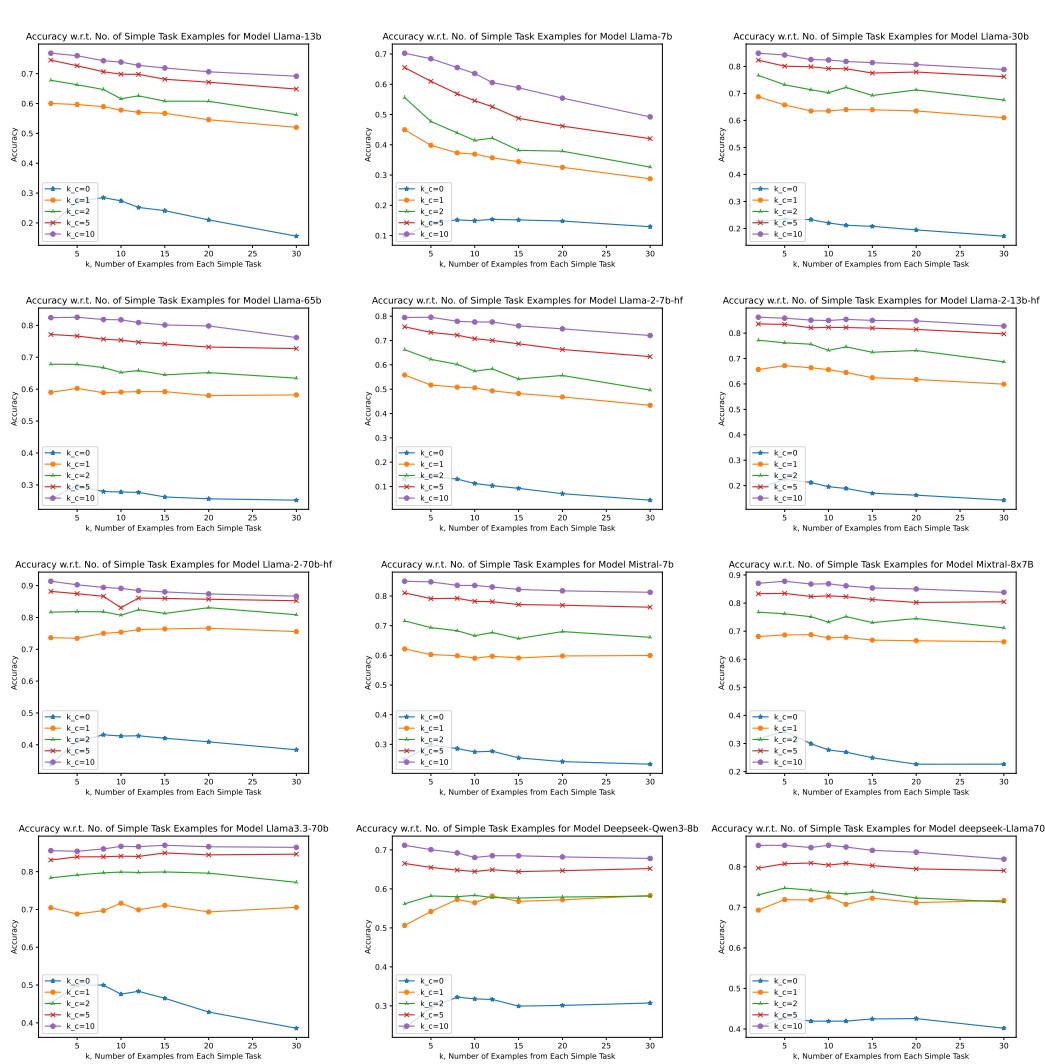

Figure 9: The effect of in-context simple task examples for each model and $k_c$, averaged over tasks.

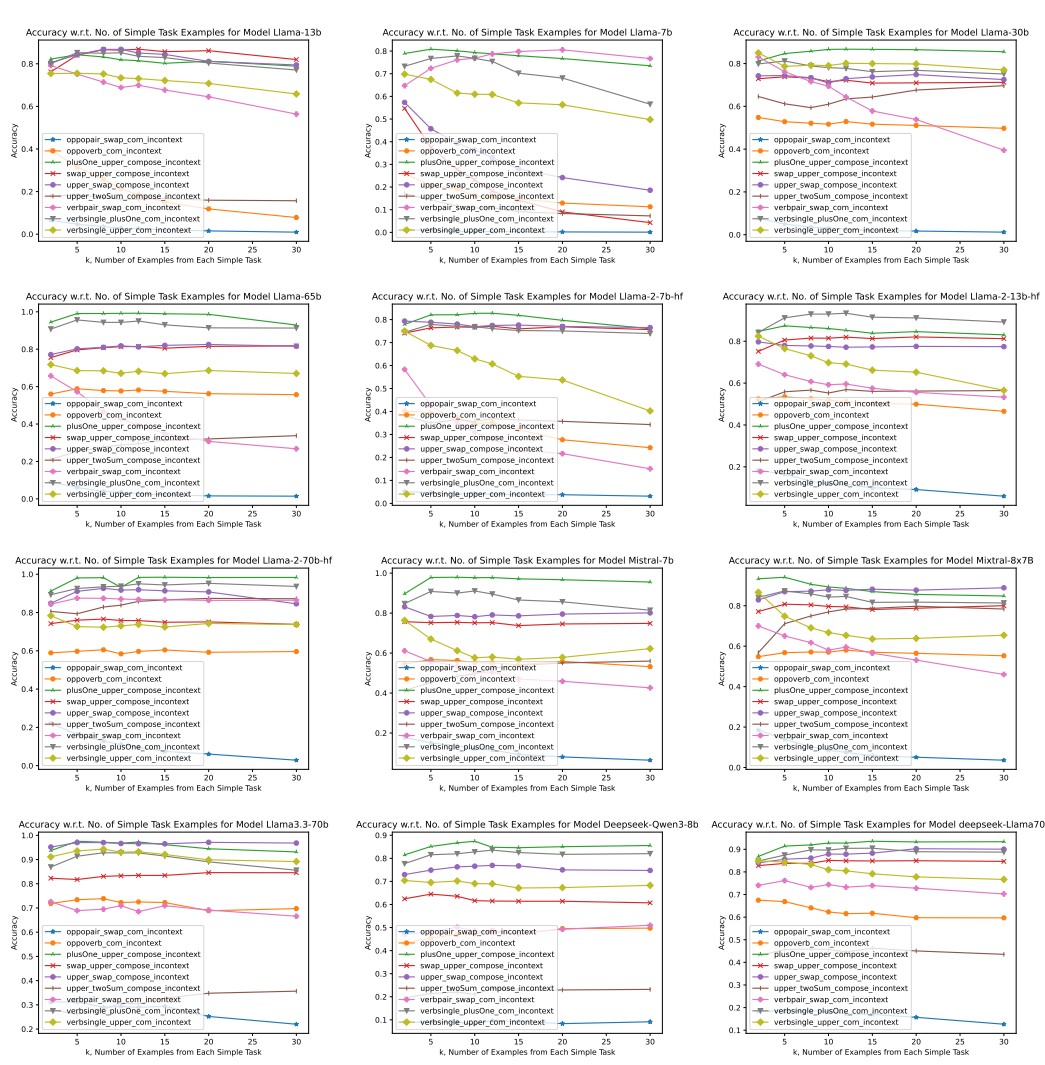

Figure 10: The effect of in-context simple task examples for each model and task, averaged over $k_c$.

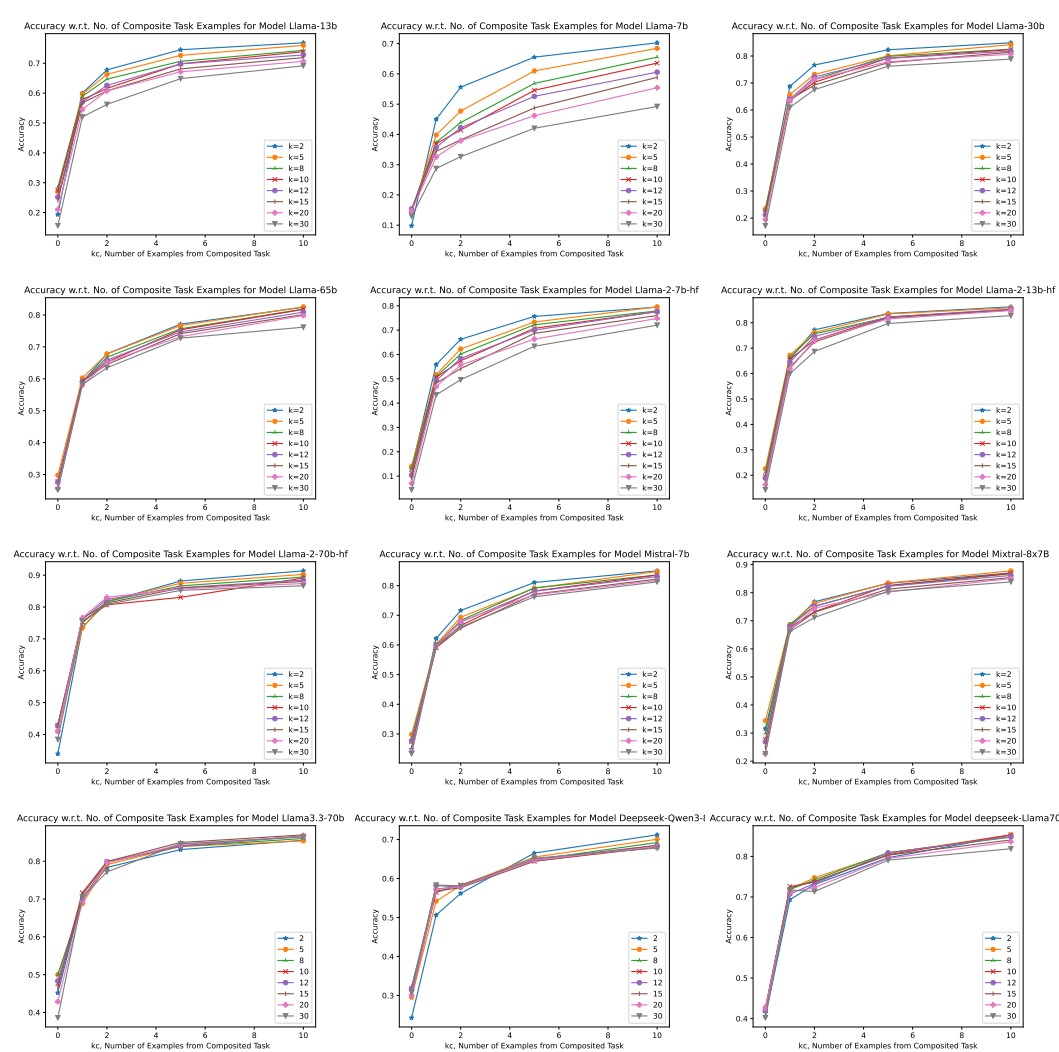

Figure 11: The effect of in-context composite task examples for each model and $k_c$, averaged over tasks.

**In-context composite task examples.** Fig. 11 shows the effect of in-context composite task examples for each $k$ and model (the reported accuracy is averaged over tasks). More precisely, we draw a subfigure for each model and draw a curve for each $k$; the $x$-axis is the number $k_c$ of examples from the composite task, and $y$-axis is the accuracy averaged over all the composite tasks.

The trend is consistent across different models and $k$'s: more composite task examples indeed help the performance on the composite queries as expected.

### B.2.1 COMPOSITE TASK EXAMPLES ARE HARMFUL FOR SIMPLE TASK QUERIES

Section 3.1 presents the result that simple task examples have an negative impact on the performance of the model on composite task queries. Here we also investigate the impact of composite task examples on the performance of the model on simple task queries.

Fig. 12 shows the change of accuracy on simple task queries when the number $k_c$ of composite task examples increases. This again confirms that the models does not correctly distinguish between composite and simple task examples for addressing the query.

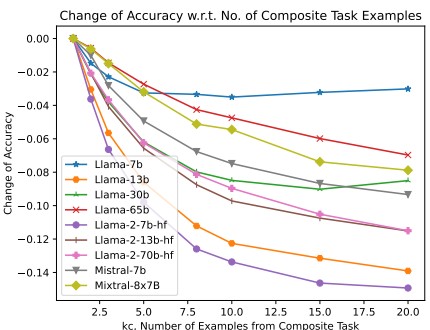

Figure 12: The effect of composite task examples on the performance on simple task queries, averaged over $k$ and tasks.

### B.3 MORE RESULTS FOR THE OUTPUT DISTRIBUTION

In the main body we examine how increasing the number $k_1$ of simple task 1 examples affects the output distribution on the opposition+swap task, with $k_2 = 10$ and $k_c = 5$. Here we examine how increasing the number $k_c$ of composite task examples affects the output. More precisely, we use the Llama-2-70B model on the opposition+swap task and use the Llama-2-7B model on the pastTense+swap task, and vary the number of composite task examples $k_c$ and fix $k_1 = k_2 = 10$.

Fig. 13 shows the results. As expected, when $k_c$ increases, the correspondence to the composite task increases while those to the simple tasks decreases. This again supports that the model does not distinguish between composite and simple task examples when utilizing them to address the query.

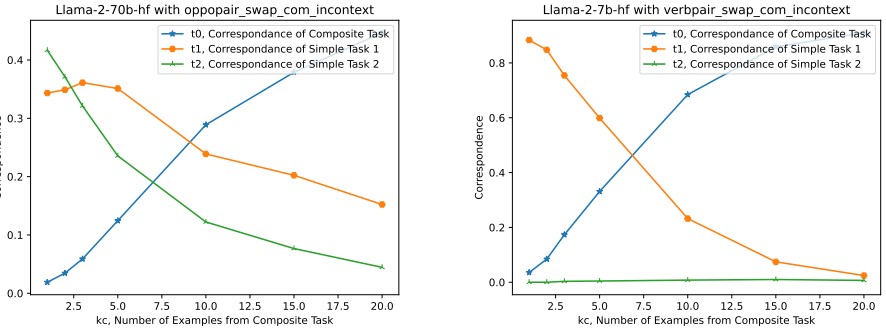

Figure 13: The output distribution for different numbers of composite task examples ($k_c$).

### B.4 DETAILED RESULTS FOR IRRELEVANT CONTENT/OPERATOR

Here we present detailed results for ablating the content or the operator in the composite task examples. We choose representative settings: $k_c = 2, 5$ for the tasks including opposition+pastTense, opposition+swap, pastTense+swap, pastTense+plusOne, and pastTense+capitalization. We report the accuracy average over the tasks (and random shuffling).

Fig. 14 shows that after ablating the content, the trend of performance decreasing with larger $k$ is roughly the same as before. This suggests that the content may not be the main factor here. Fig. 15 shows that after ablating the operator, the negative impact of larger $k$ is not as significant. This suggests that the operator may play an important role in how the model utilizes the examples.

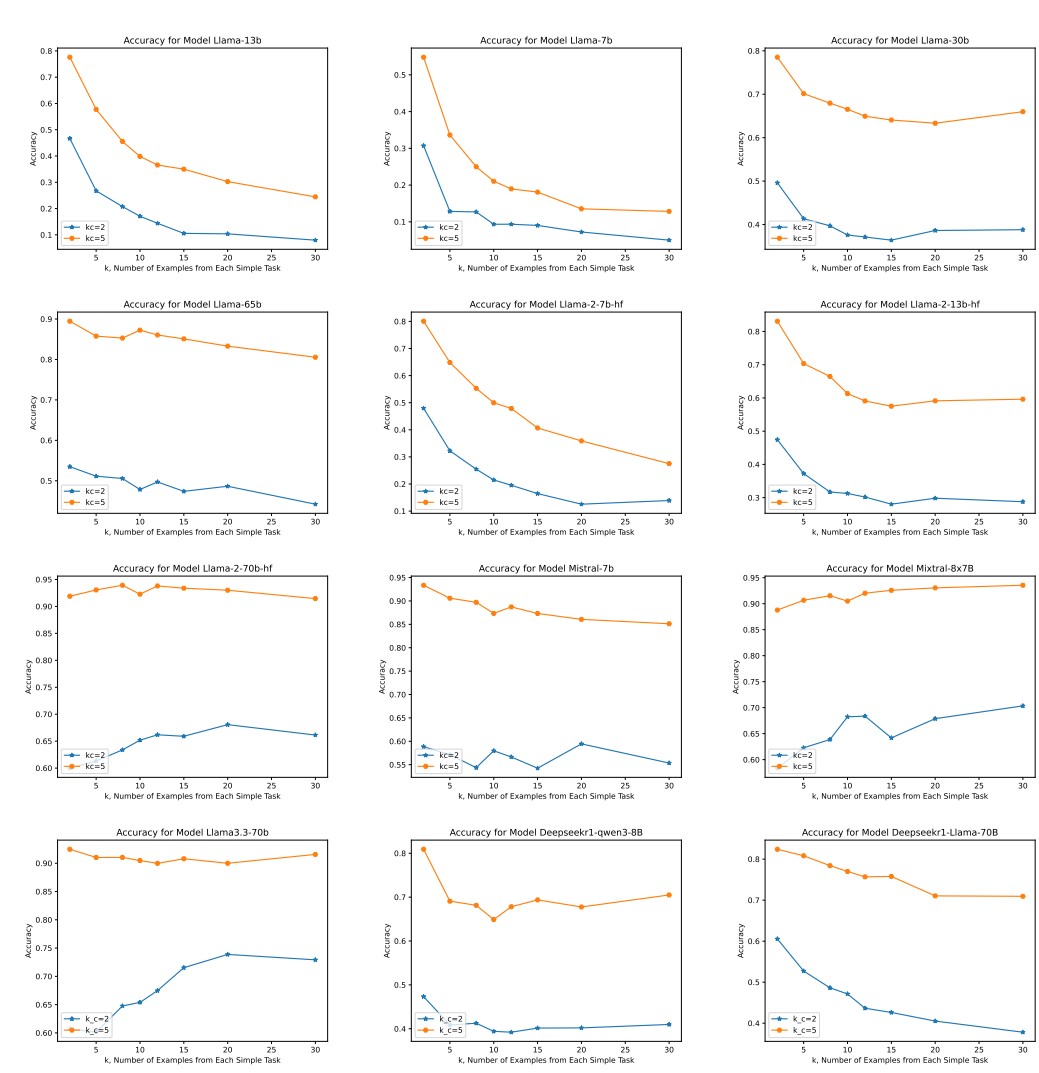

Figure 14: Results after ablating the content in the composite task examples.

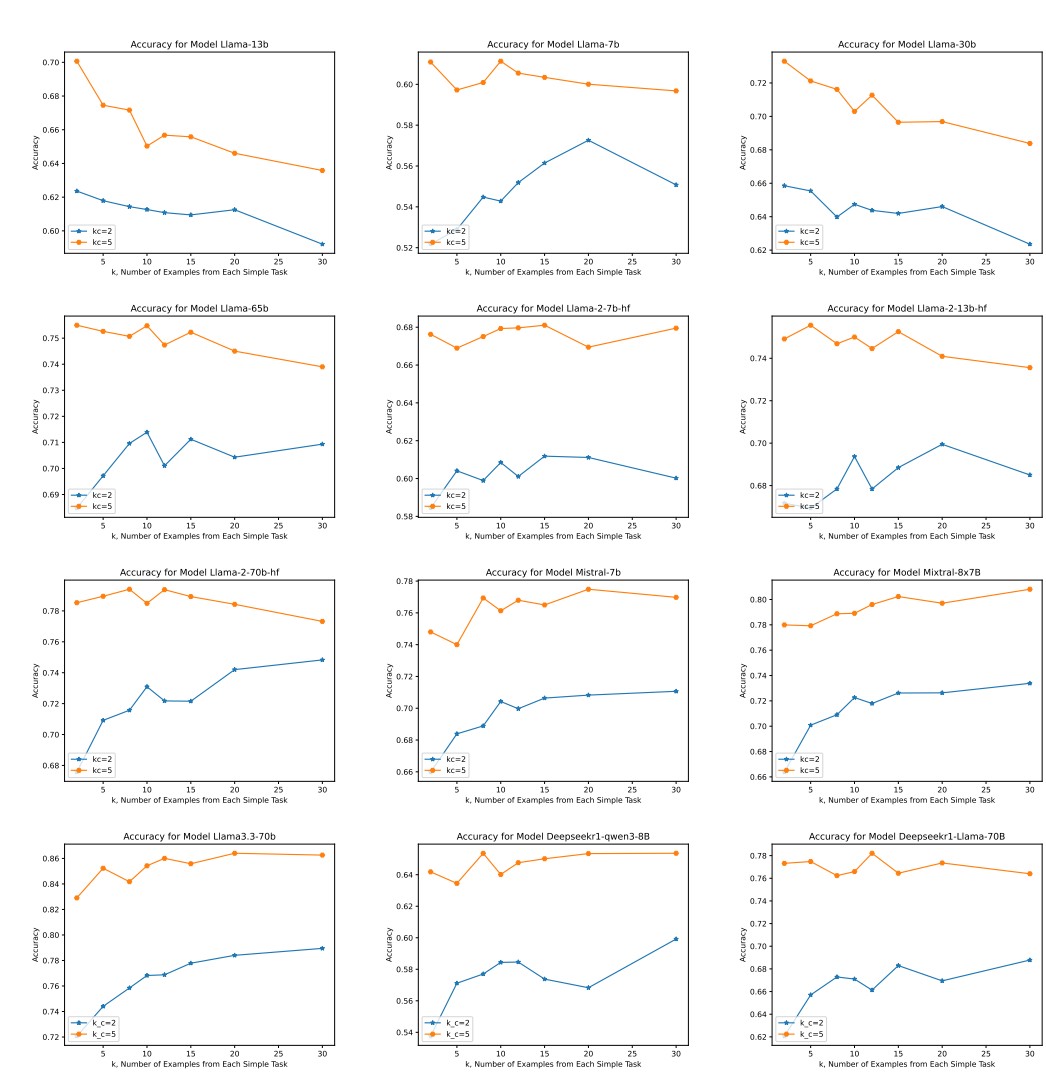

Figure 15: Results after ablating the operators in the composite task examples.

## B.5 DETAILED RESULTS FOR INNER ATTENTION

### B.5.1 MORE RESULTS FOR SIMILARITIES OF ATTENTIONS ON UNSUCCESSFUL QUERIES

In the main body, we present the similarities of attention for the opposition+swap task. In particular, we choose the opposition+swap task, fix a context, and randomly generate the queries (100 simple task queries and 100 composite task queries). The results there show that the similarities among simple and composite queries are high, suggesting the model does not distinguish the two kinds of tasks.

Here, we include some statistics about the results to further confirm the similarity.

**Average similarities and standard deviations.** We compute these average statistics to provide quantification of the similarities. There are three types of pairs of queries: simple-simple, simple-composite, composite-composite. For each type, we compute the average/standard deviation of the similarities between the attentions, and present them in Table 5. The results demonstrate that the attentions for simple or composite tasks are quite similar.

| Layer | 1 | 10 | 15 | 17 |
|---|---|---|---|---|
| Composite-Simple | $0.9997 \pm 0.0002$ | $0.9984 \pm 0.0023$ | $0.9935 \pm 0.0104$ | $0.9904 \pm 0.0142$ |
| Composite-Composite | $0.9998 \pm 0.0002$ | $0.9988 \pm 0.0019$ | $0.9950 \pm 0.0084$ | $0.9922 \pm 0.0125$ |
| Simple-Simple | $0.9998 \pm 0.0002$ | $0.9986 \pm 0.0021$ | $0.9948 \pm 0.0089$ | $0.9920 \pm 0.0125$ |
| Layer | 19 | 25 | 30 | 32 |
| Composite-Simple | $0.9876 \pm 0.0169$ | $0.9841 \pm 0.0185$ | $0.9836 \pm 0.0173$ | $0.9826 \pm 0.0178$ |
| Composite-Composite | $0.9897 \pm 0.0148$ | $0.9865 \pm 0.0160$ | $0.9860 \pm 0.0150$ | $0.9853 \pm 0.0153$ |
| Simple-Simple | $0.9895 \pm 0.0169$ | $0.9861 \pm 0.0169$ | $0.9853 \pm 0.0161$ | $0.9843 \pm 0.0169$ |

Table 5: The average and standard deviations of the attention similarities between different groups of queries for the opposition+swap task, which has low accuracy.

**Distributions of Entropy Values of the Attentions.** We further compute the entropy distribution of the attentions: for a query and a fixed layer, we extract the attentions of each head from the query to the tokens in the context, compute the entropy for each head, and then plot the histogram of the entropy for different heads and queries. The results are shown in Figure 16. The results show that the entropy distributions of the attentions for composite queries are similar to those for simple queries.

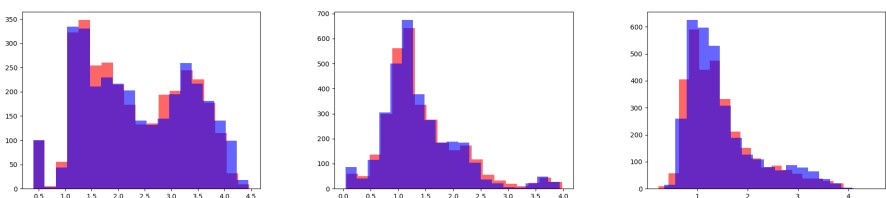

Figure 16: Entropy distributions of the attentions on 100 composite queries (red) and 100 simple queries (blue) for the opposition+swap task. Attentions are from Layer 15, 17, and 19 of Mistral-7B.

### B.5.2 DISSIMILARITIES OF ATTENTIONS ON SUCCESSFUL QUERIES

In this section, we present some additional results for the case where the model succeeds in solving the query, i.e., similarities of the attention on queries that have high accuracy. We choose the task opposition+pastTense (which has high accuracy), fixed a context, and randomly generate 100 simple task queries and 100 composite task queries.

Fig. 17 shows that for such a case, the model indeed has different patterns of attention for the two kinds of queries: simple task and composite task queries. This means that in order to achieve high accuracy, it is important for the model to distinguish the two kinds of queries. Table 6 shows the average similarities and Fig. 18 shows the distributions of the entropy values.

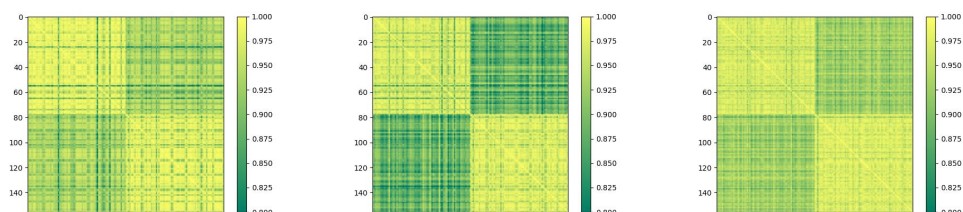

Figure 17: Similarities of the attentions between 100 simple queries (first 100 rows/columns) and 100 composite queries (last 100 rows/columns). Attentions are from Layer 15, 17, 19 of MistralAI-7b. The composite queries are selected among high accuracy ones (more precisely, from the opposition+pastTense task).

| Layer | 1 | 10 | 15 | 17 |
|---|---|---|---|---|
| Composite-Simple | $0.9991 \pm 0.0004$ | $0.9979 \pm 0.0035$ | $0.9889 \pm 0.0226$ | $0.9754 \pm 0.0442$ |
| Composite-Composite | $0.9999 \pm 0.0001$ | $0.9986 \pm 0.0034$ | $0.9938 \pm 0.0164$ | $0.9900 \pm 0.0209$ |
| Simple-Simple | $0.9998 \pm 0.0002$ | $0.9986 \pm 0.0029$ | $0.9936 \pm 0.0140$ | $0.9910 \pm 0.0165$ |

| Layer | 19 | 25 | 30 | 32 |
|---|---|---|---|---|
| Composite-Simple | $0.9715 \pm 0.0440$ | $0.9700 \pm 0.0400$ | $0.9711 \pm 0.0368$ | $0.9707 \pm 0.0360$ |
| Composite-Composite | $0.9881 \pm 0.0212$ | $0.9854 \pm 0.0204$ | $0.9850 \pm 0.0189$ | $0.9843 \pm 0.0188$ |
| Simple-Simple | $0.9893 \pm 0.0171$ | $0.9869 \pm 0.0171$ | $0.9863 \pm 0.0161$ | $0.9858 \pm 0.0160$ |

Table 6: The average and standard deviations of the attention similarities between different groups of queries for the opposition+pastTense task, which has high accuracy.

### B.5.3 RESULTS FOR AVERAGE ATTENTION FROM THE QUERY

Here we investigate the average attention from the tokens in the query to the tokens in different groups of the in-context examples. The prompt tokens are in four groups: the composite task, task 1, task 2, and the query. We compute the average attention from a token in the query to a token in some other group, to inspect where the model pays more attention when solving the query.

Fig. 19 shows that on these tasks, the same phenomenon is observed: roughly the same order of attention is paid from the query to the three different groups of examples. While this observation alone does not rule out the possibility that the model makes clever use of different groups of examples but in different ways, the observations in the other experiments above suggest that is unlikely. So combined with the observations in the other experiments, the results here suggest the model may not be able to allocate proper attention to the three groups of examples.

### B.6 DETAILED RESULTS ABOUT COT

In this section, we present detailed results for naïve applying Chain-of-Thought on the composite task examples.

Fig. 20 shows the effect of in-context simple task examples for each $k_c$ and model (the reported accuracy is averaged over tasks). More precisely, we draw a subfigure for each model and draw a curve for each $k_c$; the $x$-axis is the number $k$ of examples from each simple task, and $y$-axis is the accuracy averaged over all the composite tasks. Note that when $k_c = 0$ there are no composite task examples and thus it is meaningless to apply CoT, so we ignore this case. From the detailed results, we can see that applying CoT naïvely does not improve the accuracy much. It does not reduce the negative impact of more simple task examples either, which suggests that even with CoT composite task examples the model still cannot utilize the examples from simple tasks properly.

### B.7 THE EXPERIMENT BEFORE EXPCOT: ADDING TAGS TO SAMPLES

A natural idea to improve the composition performance is to let the model know explicitly which task each in-context example is from. This can potentially help the model distinguish between

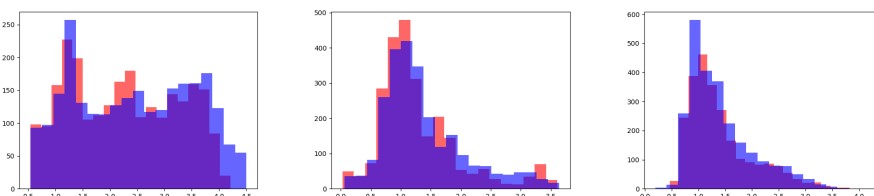

Figure 18: Entropy distributions of the attentions on 100 composite queries (red) and 100 simple queries (blue) for the opposition+pastTense task. Attentions are from Layer 15, 17, and 19 of Mistral-7B.

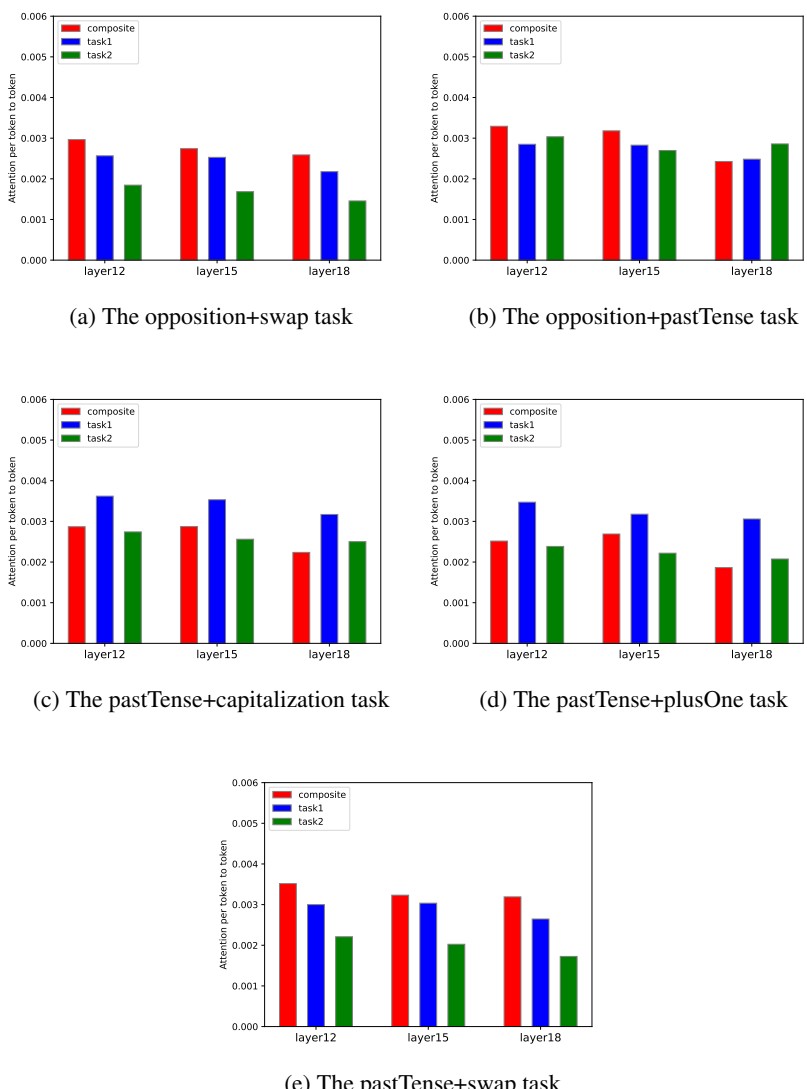

(a) The opposition+swap task

(b) The opposition+pastTense task

(c) The pastTense+capitalization task

(d) The pastTense+plusOne task

(e) The pastTense+swap task

Figure 19: Average attention from the composite task query to different groups of in-context examples.

simple/composition task examples and make better use of them. We add tags "simple1" "simple2" or "composite" to each in-context example (and the query), and rerun the experiments with $k_c = 5$.

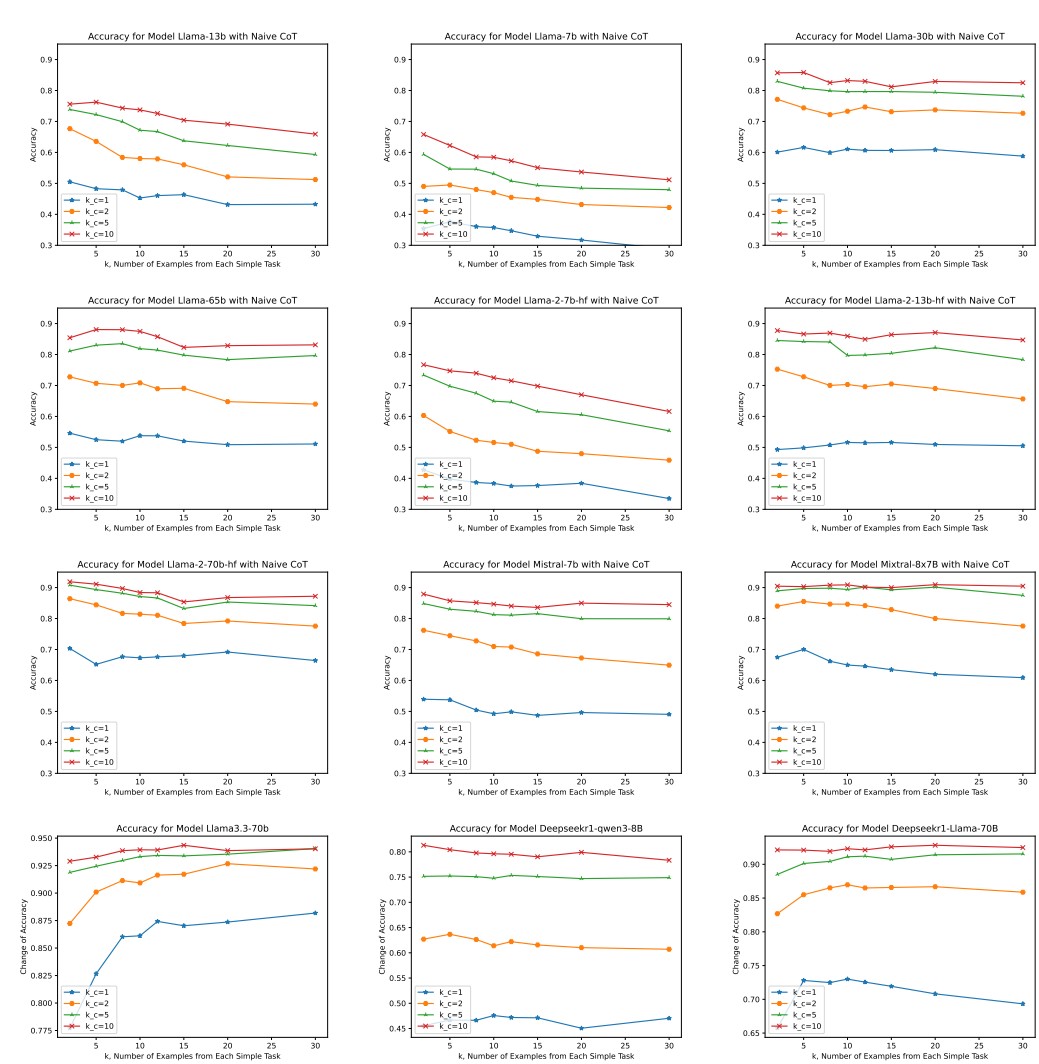

Figure 20: The effect of the in-context examples from composite tasks with naïve Chain-of-Thoughts.

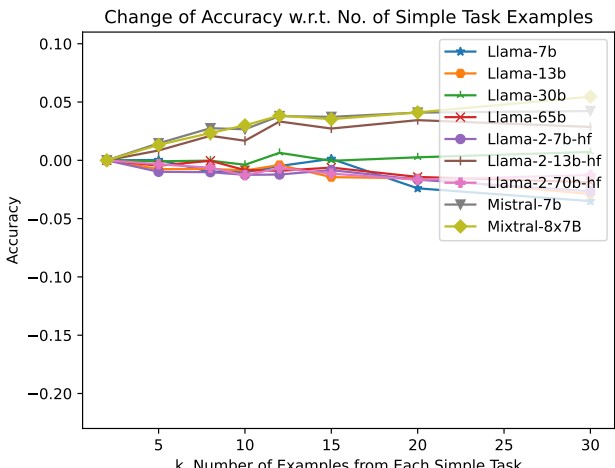

Figure 21: The change of accuracy with the number of examples from each simple task, where tags were added to each sample.

**Result.** The results show that adding tags indeed helps improve accuracy, but it does not help eliminate the negative impact of simpler task examples.

Table 7 presents the accuracy averaged over all tasks and all $k$ values, comparing the case without tags and the case with tags. It shows that adding tags indeed increases the accuracy by 1-5 percent. The result is consistent with our theoretical insight. Adding tags can help the model distinguish between simple task examples from composite ones, thus avoiding the confusion we observed in our experiments and improving the accuracy.

However, this does not help the model align the simple task examples with proper steps in the CoT. As a result, the model still cannot exploit the simple task examples effectively, and the negative impact of more simple task examples still exists as shown in Figure 21.

|  | Llama-7B | Llama-13B | Llama-30B | Llama-65B | Llama2-7B | Llama2-13B | Llama2-70B | Mistral-7B | Mistral-8x7B |
|---|---|---|---|---|---|---|---|---|---|
| No Tags | 53.4 | 69.7 | 79.0 | 74.9 | 70.0 | 82.1 | 86.0 | 78.2 | 82.0 |
| Have Tags | 64.8 | 72.1 | 81.4 | 75.7 | 76.4 | 82.2 | 87.0 | 82.3 | 84.0 |

Table 7: The accuracy (%) averaged over tasks and $k$ ($k_c = 5$), for without tags and with tags.

## B.8 DETAILED RESULTS ABOUT EXPCOT

In this section, we present detailed results for our ExpCoT method.

Fig. 22 shows the effect of in-context simple task examples for each $k_c$ and model (the reported accuracy is averaged over tasks). More precisely, we draw a subfigure for each model and draw a curve for each $k_c$; the $x$-axis is the number $k$ of examples from each simple task, and $y$-axis is the accuracy averaged over all the composite tasks. We ignore $k_c = 0$ where there is no composite task examples and thus it is meaningless to apply our method. From the detailed results, we observe that our method yields significant accuracy improvements across models and mitigates the negative impact of simple task examples. While vanilla settings show performance degradation with additional simple examples, our approach enables most models to benefit from these examples, with only minor negative effects remaining in smaller models. These findings demonstrate that our method enhances the models' ability to effectively utilize examples for in-context composition.

### B.8.1 FAILURE CASE ANALYSIS

We conducted a detailed error analysis to characterize the failure modes of ExpCoT.

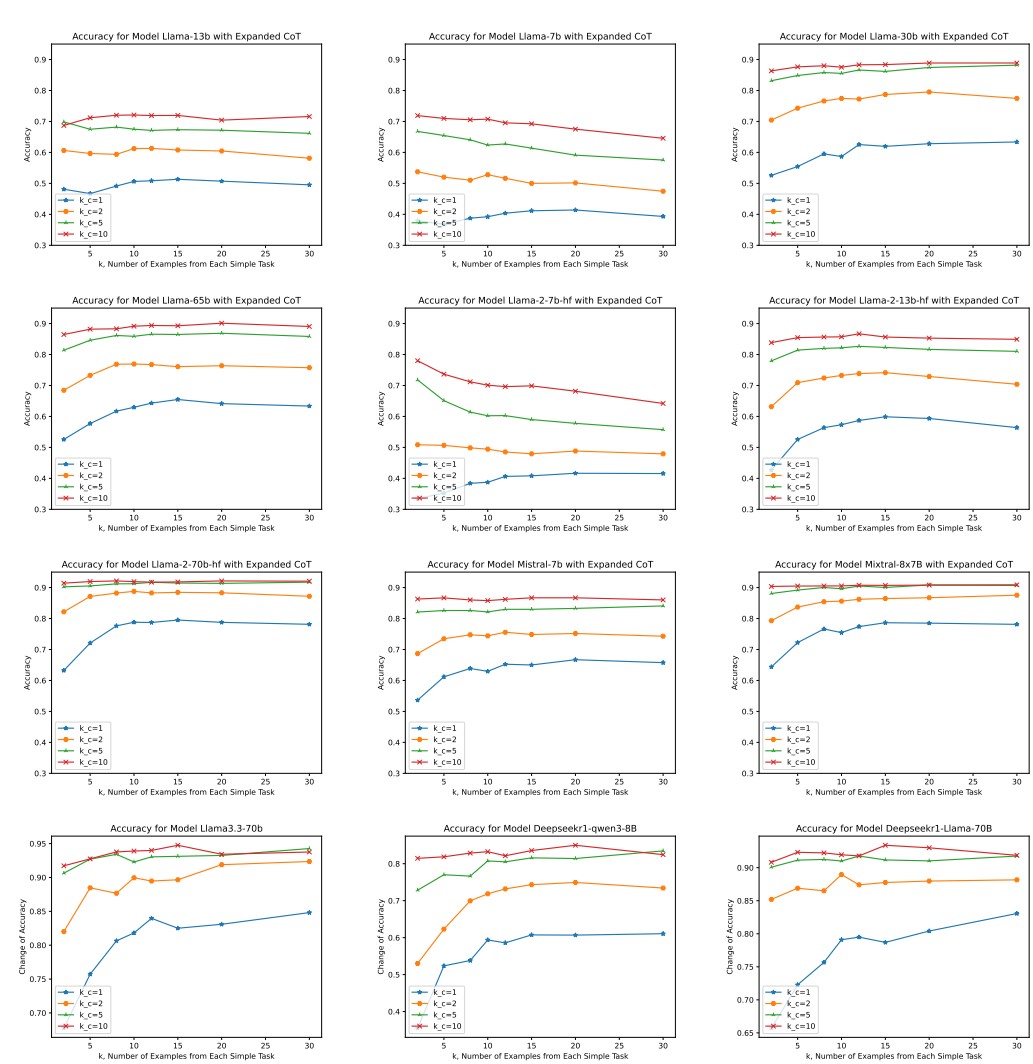

Figure 22: The effect of the in-context examples from composite tasks with ExpCoT.

We first classify the outputs into the following types:

- **Correct**: The output is correct. For example, the output is `step1: * Morning Live # ->` `step2: Evening Die # -> step3: Die Evening`. Here `*` denotes opposition and `#` denotes swap.

- **Answer_simple1**: The output only performs simple task 1. For example, `step1: * Hire` `Lend # -> step2: Borrow Fire # -> step3: Borrow Fire`. Here `*` denotes opposition and `#` denotes swap.

- **Answer_simple2**: The output only performs simple task 2, e.g., `step1: * occur #` `-> step2: occur # -> step3: OCCUR`. Here `*` denotes past tense and `#` denotes capitalization.

- **Special_token**: Outputs "???" at Step 3, e.g., `step1: * Grow Present # -> step2:` `Shrink Past # -> step3: ???`. Here `*` denotes opposition and `#` denotes swap.

- **No_Step3**: Fails to generate Step 3 entirely.

- **Step3_noexecute**: Step 3 copies Step 2, e.g., `step1: 79 bake -> step2: 79 baked ->` `step3: 79 baked`. Here simple task 1 is plus one to the number and and task 2 is past tense to verb.

- **Other Faults**: Miscellaneous errors.

Table 8 shows the distributions of the outputs for different number k. The results are averaged over all tasks and all models tested with $k_c = 2$.

The key findings are: (1) The most common failure is outputting special tokens (11-13%), suggesting models recognize the need for Step 3 but cannot determine the operation. (2) Simple task confusion is rare ($<2\%$), showing ExpCoT helps recognize compositional nature. The primary remaining challenge is helping models correctly execute aligned steps.

| Output Types (%) | $k=2$ | $k=5$ | $k=8$ | $k=10$ | $k=12$ | $k=15$ | $k=20$ | $k=30$ |
|---|---|---|---|---|---|---|---|---|
| Correct | 71.12 | 72.81 | 73.74 | 73.42 | 73.31 | 73.76 | 74.40 | 72.21 |
| Answer_simple1 | 0.12 | 0.07 | 0.05 | 0.05 | 0.10 | 0.16 | 0.14 | 0.20 |
| Answer_simple2 | 1.20 | 1.32 | 1.12 | 1.49 | 1.32 | 1.71 | 1.43 | 1.90 |
| Special_token | 13.02 | 12.52 | 11.87 | 12.10 | 11.60 | 11.31 | 11.56 | 11.61 |
| No_Step3 | 0.04 | 0.02 | 0.02 | 0.03 | 0.00 | 0.00 | 0.01 | 0.06 |
| Step3_noexecute | 0.15 | 0.07 | 0.08 | 0.07 | 0.03 | 0.07 | 0.09 | 0.11 |
| Other Faults | 14.34 | 13.18 | 13.11 | 12.85 | 13.65 | 13.00 | 12.37 | 13.91 |

Table 8: The distribution of different error types under different $k$. The results are averaged over all models and tasks for $k_c = 2$.

The results are similar across different models. To show this, the following table presents the distributions for a specific model Llama2-13b-hf. (Results for other models are also similar and we include only one for brevity.)

| Output Types (%) | $k=2$ | $k=5$ | $k=8$ | $k=10$ | $k=12$ | $k=15$ | $k=20$ | $k=30$ |
|---|---|---|---|---|---|---|---|---|
| Correct | 71.64 | 79.92 | 81.05 | 80.97 | 81.53 | 81.48 | 79.76 | 77.45 |
| Answer_simple1 | 0.00 | 0.00 | 0.00 | 0.00 | 0.11 | 0.22 | 0.11 | 0.00 |
| Answer_simple2 | 1.44 | 1.07 | 1.54 | 1.41 | 1.32 | 2.23 | 1.85 | 3.31 |
| Special_token | 14.83 | 6.01 | 4.58 | 4.67 | 3.53 | 3.77 | 3.35 | 3.47 |
| No_Step3 | 0.00 | 0.00 | 0.00 | 0.00 | 0.00 | 0.00 | 0.11 | 0.11 |
| Step3_noexec | 0.14 | 0.00 | 0.00 | 0.00 | 0.13 | 0.00 | 0.51 | 0.24 |
| Other Faults | 11.95 | 13.01 | 12.83 | 12.95 | 13.37 | 12.30 | 14.30 | 15.42 |

Table 9: The distribution of different error types under different $k$. The results are averaged over all tasks on Llama2-13b-hf for $k_c = 2$.

## B.9 EXPERIMENTS ON COMPOSITIONS OF MORE SIMPLE TASKS

We added experiments on some compositions of $T$ simple tasks with $T > 2$. We tested 4 triple compositions (i.e., $T = 3$): opposition + swap + capitalization, opposition + pastTense + capitalization, pastTense + swap + adding bracelet, pastTense + capitalization + reverse.

The results are in general consistent with those for $T = 2$: (1) more examples from simple tasks may not lead to better performance on composite queries; (2) This is due to misalignment, which can be mitigated by ExpCoT, leading to improved performance. Details are as follows.

**More examples from simple tasks may not help.** Table 10 shows the changes in the accuracy when increasing $k$, the number of examples from each simple task. We still observe that increasing the number of simple task examples may lead to worse performance. Some models are less affected, but in cases, their performance still drops, e.g., for Llama2-70B, from $k = 12$ to $k = 15$, the accuracy drops from 44.05% to 42.67%.

| Model | $k = 2$ | $k = 5$ | $k = 8$ | $k = 10$ | $k = 12$ | $k = 15$ | $k = 20$ |
|---|---|---|---|---|---|---|---|
| Llama-7B | 23.54 | 22.67 | 22.95 | 19.95 | 18.95 | 19.02 | 13.97 |
| Llama-13B | 32.03 | 32.87 | 32.13 | 31.29 | 30.22 | 29.80 | 25.01 |
| Llama-30B | 35.52 | 35.23 | 31.03 | 29.69 | 28.90 | 26.48 | 25.64 |
| Llama-65B | 40.81 | 42.34 | 39.39 | 39.99 | 39.77 | 39.67 | 39.81 |
| Llama2-7B | 29.25 | 30.14 | 30.31 | 31.93 | 32.22 | 33.01 | 28.84 |
| Llama2-13b | 39.02 | 42.50 | 42.34 | 41.76 | 40.37 | 42.64 | 41.31 |
| Llama2-70B | 42.67 | 42.93 | 43.09 | 43.16 | 44.05 | 42.67 | 45.07 |
| Mistral-7B | 42.31 | 44.23 | 44.42 | 44.95 | 43.78 | 45.57 | 45.03 |
| Mistral-8x7B | 43.45 | 43.31 | 43.00 | 42.91 | 43.65 | 42.88 | 44.67 |

Table 10: Average accuracy (in %) on the 4 triple composition tasks for different $k$.

**ExpCoT improves the performance.** We also apply ExpCoT for these compositions. Table 11 shows the performance without ExpCoT v.s. with ExpCoT (using $k_c = 5$ examples from the composition task and $k = 15$ examples from each simple task). Similar to the $T = 2$ cases, ExpCoT consistently improves the accuracy by a large margin, e.g., 25% for Mistral-8x7B. This shows that in the vanilla setting the model indeed suffers from misalignment, while ExpCoT provides hints for the model to better align skills with steps in the composition, and thus improves the performance.

| | L-7B | L-13B | L-30B | L-65B | L2-7B | L22-13B | L2-70B | M-7B | M-8x7B |
|---|---|---|---|---|---|---|---|---|---|
| Vanilla | 19.0 | 29.8 | 26.5 | 39.7 | 33.0 | 42.6 | 42.7 | 45.6 | 42.9 |
| ExpCoT | 44.2 | 38.8 | 54.8 | 61.7 | 50.9 | 55.9 | 64.8 | 57.7 | 67.9 |

Table 11: The average accuracy (in %) on the 4 triple compositions, with $k = 15$ and $k_c = 5$. L: Llama, L2/3: Llama2/3.3, M: Mistral.

## B.10 EXPERIMENTS ON MORE MODELS

| | $k = 2$ | $k = 5$ | $k = 8$ | $k = 10$ | $k = 12$ | $k = 15$ | $k = 20$ | $k = 30$ |
|---|---|---|---|---|---|---|---|---|
| GPT4.1 (Vanilla) | 20 | 28 | 34 | 33 | 25 | 25 | 27 | 22 |
| GPT4.1 (ExpCoT) | 65 | 64 | 73 | 69 | 59 | 66 | 69 | 66 |
| Gemini2.5 (Vanilla) | 42 | 40 | 53 | 56 | 53 | 56 | 48 | 57 |
| Gemini2.5 (ExpCoT) | 69 | 70 | 72 | 75 | 70 | 75 | 74 | 69 |

Table 12: Accuracy (in %) of GPT4.1 and Gemini2.5 on the opposition+swap task for different $k$.

We additionally test some closed models like GPT4.1 and Gemini2.5. We would like to note that our work focuses on open-source models for which we can systematically control model sizes, families, and prompting variants, and even inspect the internal attentions. The purpose is to identify the key

factors in a transparent setting. On the other hand, results from some closed models provide further support of our analysis.

Table 12 shows the results of GPT4.1 and Gemini2.5 on the composition task opposition+swap, with $k$ examples from each simple task and $k_c = 5$ examples from the composite task (the same setting as in Figure 1).

- We observe that for GPT4.1 without ExpCoT (vanilla setting), initially (from k=2 to $8$), increasing $k$ leads to better performance, while later increasing $k$ leads to worse performance. Gemini2.5 also shows a similar pattern. This shows that the model still cannot fully recognize the composition structure and align examples with the correct steps in the composition, but it may partially exploit the examples: at first, the benefit of exploiting the examples to infer the skills outweighs the negative impact of misalignment, but later the negative impact dominates.

- ExpCoT improves the performance significantly. This shows that the hints on aligning the skills and composition steps can mitigate the misalignment issue and thus improve the accuracy.

## C  DETAILS OF THEORETICAL ANALYSIS

First recall the theoretical setup. A sequence-to-sequence task on a finite vocabulary of tokens $\Sigma$ is associated with an input distribution $\mathcal{D}$ over the input $\mathcal{X} \subseteq \Sigma^*$, and a target function $f : \Sigma^* \to \Sigma^*$ where $f \in \mathcal{H}$ for some model class $\mathcal{H}$. A composite task with the target function $f \in \mathcal{H}^T$ can consist of $T$ steps $f_1, f_2, \ldots, f_T \in \mathcal{H}$, such that $f(x) = f_T \circ \cdots \circ f_2 \circ f_1(x)$. For simplicity, assume $\mathcal{H}$ is finite, and it includes the identity mapping so that $\mathcal{H} \subseteq \mathcal{H}^T$.

Now we present the detailed proofs for our theoretical results.

Consider the case when $k_c$ composite task examples $\mathcal{S}_0 = \{(x_i, y_i) : i \in [k_c]\}$ are given, where $x_i$ are i.i.d. from $\mathcal{D}$ and $y_i = f(x_i)$ for some $f \in \mathcal{H}^T$.

**Proposition 3** (Restatement of Proposition 1). *There exists a learning rule $\mathcal{M} : (\mathcal{X} \times \Sigma^*)^* \to \Sigma^{\mathcal{X}}$ such that for any distribution $\mathcal{D}$ over $\mathcal{X}$ and any $f \in \mathcal{H}^T$, for every $0 < \delta < 1$, we have with probability at least $1 - \delta$ over $\mathcal{S}_0$,*

$$\Pr_{x \sim \mathcal{D}}[\mathcal{M}(\mathcal{S}_0)(x) \neq f(x)] \leq \frac{1}{k_c}\left(T \ln |\mathcal{H}| + \ln\left(\frac{1}{\delta}\right)\right).$$

*Proof.* The result follows a standard argument of consistent models from a finite hypothesis class. Let $\mathcal{M}(\mathcal{S}_c)$ output a consistent model:

$$\mathcal{M}(\mathcal{S}_c) \in \{g \in \mathcal{H}^T : g(x) = y, \forall (x, y) \in \mathcal{S}_c\}. \tag{2}$$

Let $d(f, g) = \Pr_{x \sim \mathcal{D}}[g(x) \neq f(x)]$ denote the difference between $f$ and $g$. For a fixed $g$ with $d(f, g) > \epsilon$, we have

$$\Pr[g(x) = y, \forall (x, y) \in \mathcal{S}_c] \leq (1 - \epsilon)^{k_c}. \tag{3}$$

So

$$\Pr[\exists g \in \mathcal{H}^T, g(x) = y, \forall (x, y) \in \mathcal{S}_c] \leq |\mathcal{H}^T|(1 - \epsilon)^{k_c}. \tag{4}$$

Letting the right-hand side bounded by $\delta$ leads to the result. $\square$

Next we consider examples from multiple tasks. Recall that $\mathcal{S}_t$ is a set of $k_t$ examples from the $t$-th task $(\mathcal{D}_t, f_t)(0 \leq t \leq T)$. We say $\mathcal{M}$ is *focusing* if its expected error on the 0-th task is no worse than that on any other task, i.e., for any $j \in [T]$,

$$\mathcal{L}_0(\mathcal{M}; (\mathcal{D}_t, f_t)_{t=0}^T) \leq \mathcal{L}_j(\mathcal{M}; (\mathcal{D}_t, f_t)_{t=0}^T) \tag{5}$$

where $\mathcal{L}_j(\mathcal{M}; (\mathcal{D}_t, f_t)_{t=0}^T) := \mathbb{E}_{\mathcal{S}_t \sim (\mathcal{D}_t, f_t), 0 \leq t \leq T} \Pr_{x \sim \mathcal{D}_j}[\mathcal{M}(\mathcal{S}_0; \mathcal{S}_1, \ldots, \mathcal{S}_T)(x) \neq f_j(x)]$ is the expected error of $\mathcal{M}$ on the $j$-th task. And we say that $\mathcal{M}$ does *not distinguish examples from different tasks*, if it is symmetric w.r.t. the data sets $\mathcal{S}_t$'s, i.e., for any permutation $\sigma$ on $\{0, 1, \ldots, T\}$, the distribution of $\mathcal{M}(\mathcal{S}_{\sigma(0)}; \mathcal{S}_{\sigma(1)}, \ldots, \mathcal{S}_{\sigma(T)})$ is the same as that of $\mathcal{M}(\mathcal{S}_0; \mathcal{S}_1, \ldots, \mathcal{S}_T)$. Then we have:

**Proposition 4** (Restatement of Proposition 2). *Suppose there exist $g_1, \ldots, g_T \in \mathcal{H}$ with pairwise difference at least $\Delta$ for some $\mathcal{D}$, i.e., $\min_{i \neq j} \Pr_{x \sim \mathcal{D}}[g_i(x) \neq g_j(x)] \geq \Delta$. For any $\mathcal{M}$ that is focusing but does not distinguish between examples from different tasks, there exist $f_1, \ldots, f_T \in \mathcal{H}$, $f_0 = f_T \circ \cdots f_2 \circ f_1$, and $\mathcal{D}_t(0 \leq t \leq T)$'s, such that $\mathbb{E}_{\{\mathcal{S}_t\}} \Pr_{x \sim \mathcal{D}_0}[\mathcal{M}(\mathcal{S}_0; \mathcal{S}_1, \ldots, \mathcal{S}_T)(x) \neq f_0(x)] = \Omega(\Delta)$.*

*Proof.* Let $f_t = g_t$ and $\mathcal{D}_t = \mathcal{D}$ for $t \in [T]$. For datasets $\mathcal{V}_0, \mathcal{V}_1, \ldots, \mathcal{V}_T$ and a task $(\mathcal{D}_j, f_j)$, define

$$\mathcal{L}(\mathcal{M}; \mathcal{V}_0, \mathcal{V}_1, \ldots, \mathcal{V}_T; \mathcal{D}_j, f_j) := \Pr_{x \sim \mathcal{D}_j}[\mathcal{M}(\mathcal{V}_0; \mathcal{V}_1, \ldots, \mathcal{V}_T)(x) \neq f_j(x)]. \tag{6}$$

Consider a uniform distribution $\mathcal{U}$ over permutations on $\{0, 1, \ldots, T\}$. We have

$$\mathbb{E}_{\sigma \sim \mathcal{U}} \mathbb{E}_{\{\mathcal{S}_t\}} \mathcal{L}(\mathcal{M}; \mathcal{S}_{\sigma(0)}, \mathcal{S}_{\sigma(1)}, \ldots, \mathcal{S}_{\sigma(T)}; \mathcal{D}_{\sigma(0)}, f_{\sigma(0)}) \tag{7}$$

$$= \mathbb{E}_{\sigma \sim \mathcal{U}} \mathbb{E}_{\{\mathcal{S}_t\}} \mathcal{L}(\mathcal{M}; \mathcal{S}_0, \mathcal{S}_1, \ldots, \mathcal{S}_T; \mathcal{D}_{\sigma(0)}, f_{\sigma(0)}) \tag{8}$$

$$= \frac{1}{T+1} \sum_{t=0}^T \mathbb{E}_{\{\mathcal{S}_t\}} \mathcal{L}(\mathcal{M}; \mathcal{S}_0, \mathcal{S}_1, \ldots, \mathcal{S}_T; \mathcal{D}, f_t) \tag{9}$$

where the first equation comes from the assumption that $\mathcal{M}$ does not distinguish between examples from different tasks, and the second equation is because $\sigma$ is uniformly at random. We have the following triangle inequality about the error by definition:

**Claim 1.** *For any* $i, j \in [T]$,

$$\mathcal{L}(\mathcal{M}; \mathcal{S}_0, \mathcal{S}_1, \ldots, \mathcal{S}_T; \mathcal{D}, f_i) + \mathcal{L}(\mathcal{M}; \mathcal{S}_0, \mathcal{S}_1, \ldots, \mathcal{S}_T; \mathcal{D}, f_j) \geq \Pr_{x \sim \mathcal{D}}[f_i(x) \neq f_j(x)] \tag{10}$$

Then

$$\mathbb{E}_{\sigma \sim \mathcal{U}} \mathbb{E}_{\{\mathcal{S}_t\}} \mathcal{L}(\mathcal{M}; \mathcal{S}_{\sigma(0)}, \mathcal{S}_{\sigma(1)}, \ldots, \mathcal{S}_{\sigma(T)}; \mathcal{D}_{\sigma(0)}, f_{\sigma(0)}) \tag{11}$$

$$= \frac{1}{T+1} \sum_{t=0}^{T} \mathbb{E}_{\{\mathcal{S}_t\}} \mathcal{L}(\mathcal{M}; \mathcal{S}_0, \mathcal{S}_1, \ldots, \mathcal{S}_T; \mathcal{D}, f_t) \tag{12}$$

$$\geq \frac{1}{T(T+1)} \sum_{t=1}^{T} T \mathbb{E}_{\{\mathcal{S}_t\}} \mathcal{L}(\mathcal{M}; \mathcal{S}_0, \mathcal{S}_1, \ldots, \mathcal{S}_T; \mathcal{D}, f_t) \tag{13}$$

$$= \frac{1}{T(T+1)} \sum_{i=1}^{T} \sum_{j=1}^{T} \mathbb{E}_{\{\mathcal{S}_t\}} \mathcal{L}(\mathcal{M}; \mathcal{S}_0, \mathcal{S}_1, \ldots, \mathcal{S}_T; \mathcal{D}, f_i) + \mathbb{E}_{\{\mathcal{S}_t\}} \mathcal{L}(\mathcal{M}; \mathcal{S}_0, \mathcal{S}_1, \ldots, \mathcal{S}_T; \mathcal{D}, f_j)$$

$$\tag{14}$$

$$\geq \frac{1}{T(T+1)} \sum_{i=1}^{T} \sum_{j=1}^{T} \Pr_{x \sim \mathcal{D}}[f_i(x) \neq f_j(x)] \tag{15}$$

$$\geq \frac{T-1}{T+1} \Delta. \tag{16}$$

On the other hand, since $\mathcal{M}$ is focusing,

$$\mathbb{E}_{\{\mathcal{S}_t\}} \mathcal{L}(\mathcal{M}; \mathcal{S}_{\sigma(0)}, \mathcal{S}_{\sigma(1)}, \ldots, \mathcal{S}_{\sigma(T)}; \mathcal{D}_{\sigma(0)}, f_{\sigma(0)}) \tag{17}$$

$$= \mathcal{L}_0(\mathcal{M}; (\mathcal{D}_{\sigma(t)}, f_{\sigma(t)})_{t=0}^{T}) \tag{18}$$

$$\leq \mathcal{L}_{\sigma^{-1}(0)}(\mathcal{M}; (\mathcal{D}_{\sigma(t)}, f_{\sigma(t)})_{t=0}^{T}) \tag{19}$$

$$= \mathbb{E}_{\{\mathcal{S}_t\}} \mathcal{L}(\mathcal{M}; \mathcal{S}_0, \mathcal{S}_{\sigma(1)}, \ldots, \mathcal{S}_{\sigma(T)}; \mathcal{D}_{\sigma(\sigma^{-1}(0))}, f_{\sigma(\sigma^{-1}(0))}) \tag{20}$$

$$= \mathbb{E}_{\{\mathcal{S}_t\}} \mathcal{L}(\mathcal{M}; \mathcal{S}_0, \mathcal{S}_1, \ldots, \mathcal{S}_T; \mathcal{D}_0, f_0) \tag{21}$$

$$= \mathbb{E}_{\{\mathcal{S}_t\}} \Pr_{x \sim \mathcal{D}_0}[\mathcal{M}(\mathcal{S}_0; \mathcal{S}_1, \ldots, \mathcal{S}_T)(x) \neq f_0(x)]. \tag{22}$$

Combining the above two inequalities, we have

$$\mathbb{E}_{\{\mathcal{S}_t\}} \Pr_{x \sim \mathcal{D}_0}[\mathcal{M}(\mathcal{S}_0; \mathcal{S}_1, \ldots, \mathcal{S}_T)(x) \neq f_0(x)] \geq \frac{T-1}{T+1} \Delta. \tag{23}$$

This finishes the proof. $\square$

**Theorem 2** (Restatement of Theorem 1). *Suppose we are given $k_t$ examples $\mathcal{S}_t$ from $(\mathcal{D}_t, f_t)$ for $t \in [T]$ and $k_c$ examples $\mathcal{S}_0$ from $(\mathcal{D}_0, f_0)$ with $f_0 = f_T \circ \ldots \circ f_2 \circ f_1$. Suppose $\mathcal{H}$ is distinguishable: for some $\epsilon_0 > 0$, for any $f \neq g \in \mathcal{H}$ and $\mathcal{D}_t(0 \leq t \leq T)$, $\Pr_{x \sim \mathcal{D}_t}[f(x) \neq g(x)] > \epsilon_0$. There exists a learning rule $\mathcal{M} : ((\mathcal{X} \times \Sigma^*)^*)^{T+1} \to \Sigma^{\mathcal{X}}$ such that for every $0 < \delta < 1$, if*

$$\max(k_c, k_t) \geq \frac{2}{\epsilon_0} \left( \ln |\mathcal{H}| + \ln \frac{T}{\delta} \right), \quad \forall t \in [T],$$

*then with probability at least $1 - \delta$ over $\{\mathcal{S}_t\}_{t=0}^{T}$, we have $\mathcal{M}(\mathcal{S}_0; \mathcal{S}_1, \ldots, \mathcal{S}_T) = f_0$.*

*Proof.* Consider each step $t$ in the composition, which can be learned by the examples from this simple task and the corresponding intermediate outputs from the composite task examples. This high-level idea is similar to that in Abedsoltan et al. (2025); Joshi et al. (2025), but our setting is quite different, e.g., we have examples for each single step of the composition.

More precisely, for learning $f_t$, we have $\mathcal{S}_t = \{(x_i, y_i) : i \in [k_t]\}$ from the simple task, and $\mathcal{S}_{0,t} = \{(z_i^t, z_i^{t+1}) : i \in [k_c]\}$ from the composite task CoT examples. Output a model consistent with all these data:

$$\hat{f}_t \in \{g \in \mathcal{H} : g(x) = y, \forall (x, y) \in \mathcal{S}_t \cup \mathcal{S}_{0,t}\}. \tag{24}$$

For a fixed $g$ with $\Pr_{x \sim \mathcal{D}_t}[g(x) \neq f_t(x)] > \epsilon_0$, we have

$$\Pr[g(x) = y, \forall (x, y) \in \mathcal{S}_t] \leq (1 - \epsilon_0)^{k_t}. \tag{25}$$

So

$$\Pr[\exists g \in \mathcal{H}^T, g(x) = y, \forall (x, y) \in \mathcal{S}_c] \leq |\mathcal{H}|(1 - \epsilon_0)^{k_t}. \tag{26}$$

Letting the right-hand side bounded by $\delta/T$, we know that if

$$k_t \geq \frac{1}{\epsilon_0}\left(\ln|\mathcal{H}| + \ln\frac{T}{\delta}\right), \tag{27}$$

then with probability at least $1 - \delta/T$, $\hat{f}_t = f_t$. A similar argument holds for the $k_c$ examples from the composite task. Taking a union bound over the $T$ steps leads to the statement. $\qquad\square$

Note that if the $t$-th simple task example input data distribution $\mathcal{D}_t$ is the same as the distribution of $f_t \circ \ldots \circ f_1(x)$ where $x \sim \mathcal{D}_0$, then the sample complexity is improved to:

$$k_c + k_t \geq \frac{1}{\epsilon_0}\left(\ln|\mathcal{H}| + \ln\frac{T}{\delta}\right). \tag{28}$$

