# OpenReview forum: "Can Language Models Compose Skills In-Context?"
_ICLR.cc/2026/Conference — Submitted to ICLR 2026_

### Official Review · Reviewer_eFa6 · 2025-10-28

**Soundness:** 3
**Presentation:** 3
**Contribution:** 3
**Rating:** 8
**Confidence:** 3

**Summary:**

The paper investigates whether LLMs can perform compositional tasks that combine basic skills demonstrated in in-context examples. The work finds that simple in-context examples can (surprisingly) hurt final performance on the compositional task. Further ablations, closer look at attention maps, and theoretical analysis support the following explanation for the observed failure mode: models generally do not recognize the composition and do not align the simple task in-context examples with the corresponding steps of the composition. Motivated by this, the authors propose ExpCOT, which converts simple examples into step-labeled COT with “missing steps”, improving performance across many open source models.

**Strengths:**

1) The paper clearly communicates the observed phenomena of interest: including simple examples in context can hurt final performance on a compositional task.
2) The paper offers a plausible explanation for the phenomena, supported by theory + experiments.
3) The work also offers a simple mitigation (via ExpCoT) for the observed failure mode.

**Weaknesses:**

1) The paper does not report performance on the simple tasks themselves under simple-only in-context prompts (or even zero-shot); it only shows how performance on simple tasks changes when compositional in-context examples are included (my apologies if I have overlooked something). Without this clean baseline, it becomes difficult to disentangle true compositional failure from insufficient competence on the underlying atomic skills. By finetuning on those atomic skills, does including the simple skills in-context still hurt performance on compositional tasks?
2) Most experiments focus on T=2 step compositions, but what about larger T? (The theory is for general, larger T).
3) (Minor typo) Paper mispells “Llama” as “Llamma”

**Questions:**

1) Could the authors comment on relation to the following work: https://arxiv.org/abs/2505.00147 (e.g., the linked work including the simple skills in context can hurt final performance on easy questions by introducing unnecessary information)

---

> ### Author Response · Authors · 2025-11-20
> **Author Response**
>
> We thank the reviewer for acknowledging that (i) the phenomenon is interesting, (ii) our theory and experiments provide some plausible explanation, (iii) ExpCoT is simple and effective.
>
> ## Weakness 1 (1): report performance on the simple tasks
> We mainly adopt synthetic tasks from [2], where simple-task performance has already been reported. In particular, Figure 2 in [2] shows that the performance on simple tasks model is typically close to perfect, and much better than the performance on composite tasks. Thanks for noting this, and we will point this out in our revised version.
>
> ## Weakness 1 (2): Results about model finetuning on atomic tasks
>
> We agree that fine-tuning on atomic skills may strengthen base competence.  However, unless the fine-tuning also teaches the model how to align simple examples with composition steps (e.g., by exposing it to step-labeled or decomposed training signals), the failure mode we identify can still arise at inference time when composition is only inferred by in-context examples.
>
> ## Weakness 2: Scenarios for T > 2
>
> See Global Response: Additional results for compositions with more than 2 simple tasks (T > 2). In summary, the results are in general consistent with those for T=2: (1) more examples from simple tasks may not lead to better performance on composite queries; (2) ExpCoT helps alignment and improves performance.
>
> ## Weakness 3: Typo
>
> We thank the reviewer for catching this. We will correct all instances of this typo in the final version.
>
> ## Question 1: Relation with [1]
>
> Thanks for pointing out the related work and we will add the citation and comparison to the revision. In summary, [1] and our work have connections and also differences, which are detailed below.
>
> **Connections:** Both [1] and our work study composition in in-context learning. Our work observed the negative impact of more examples from simple tasks. [1] also observed related phenomena (but in a somewhat different setting): adding more simple skill-based in-context examples can hurt performance, rather than help, and noted that this is due to overthinking of the small models on easy questions.
>
> **Differences:** There are some subtle and important differences between [1] and our work.
> - Task setting. In [1], the query is **self-explanatory** and contains information for the LLMs to infer what skills are needed. An example given there is “3+5*2=” where the operators + and * have been seen in pretraining (i.e., addition and multiplication skills). Frontier LLMs can infer the skills required and then pick corresponding examples from a given pool of examples. But [1] noted that such a strategy can hurt the performance of a small model on easy questions, ie, the in-context examples for simple skills can introduce unnecessary information, akin to cognitive overload. [1] thus propose an adaptive method that avoids skill-based examples and adds examples targeted to the skills missing from the model’s responses. In contrast, in our work, the query is **not** self-explanatory: the meaning of operators is not seen in the pretraining but only illustrated in the in-context examples provided. An example query is “*Eager Proud #” where the skills for the operators * and # are demonstrated in in-context examples and need to be inferred by the model from the examples at test time. We observe the more fundamental challenges of LLM, which struggles with using simple-task demonstrations to infer and execute a new composition. LLMs often apply only one sub-skill or match on operators without correctly aligning which skill should be used at which step.
>
> - Research Focus. [1] focuses on metacognition, i.e., recognizing the required skills for the query and retrieving proper examples from a given pool for the skills. It discovers the overthinking phenomenon of small models on easy questions and proposes an adaptive selection algorithm of in-context examples to address the issue. Our work focuses on the in-context composition ability, i.e., whether the model can learn and compose the skills from the given in-context examples. It aims to provide theoretical explanations of when and why the model can do the composition of skills only demonstrated in in-context examples. We thus use operators whose meaning is not seen in pretraining but only demonstrated in-context, thus forcing the model to learn and compose the skills.
>
> [1] He et al. AdaptMI: Adaptive Skill-based In-context Math Instructions for Small Language Models. COLM, 2025.
>
> [2] Xu et al. Do large language models have compositional ability? An investigation into limitations and scalability. COLM, 2024.

---

### Official Review · Reviewer_aP4i · 2025-10-31

**Soundness:** 3
**Presentation:** 3
**Contribution:** 3
**Rating:** 6
**Confidence:** 3

**Summary:**

The paper studies whether LMs can compose skills in-context. Using synthetic linguistic/logical compositions (e.g., opposition ∘ swap), the authors show an empirical finding: adding more simple-task (no composition) often hurts accuracy on composite queries, but adding composite-task helps. Analysis of outputs and attentions suggests models match on operators rather than semantics and often apply only one sub-skill. Naïve CoT doesn’t reliably fix this. A stylized theory shows that without distinguishing simple vs. composite examples, error can remain Ω(Δ); if examples are aligned to composition steps, sample needs can scale only logarithmically in T. Motivated by this, the paper proposes Expanded Chain-of-Thought (ExpCoT), which annotates steps and treats simple examples as CoT with missing steps; this improves accuracy across many open models.

**Strengths:**

Clear negative result that challenges common intuition: more shots of sub-skills can degrade composition performance; trend is shown across many models/sizes and tasks.

Careful empirical probing: shuffling to reduce order sensitivity, correspondence analysis showing models often perform just one sub-task, and operator-vs-content ablations clarifying what is attended to.

Theoretical insight: formalizes when failing to distinguish data sources leads to lower bounds, and when step-aligned evidence can provably help (log T scaling).

Simple fix method: ExpCoT is simple to implement, consistent gains over vanilla and naïve CoT.

**Weaknesses:**

Scope limited to synthetic tasks with T=2: It’s unclear whether the phenomenon and ExpCoT gains persist for (i) more realistic multi-hop QA/program induction, (ii) longer compositions (T>2), or (iii) noisy/ambiguous operators. Current tasks may overemphasize symbol/operator cues. (Datasets: 9 compositions from eight base skills.)

Model coverage & claims: Results exclude strongest closed models; conclusions about “LLMs in general” may overreach. The paper notes the resource constraint, but the core claim would be stronger with at least one strong proprietary baseline.

**Questions:**

Scaling in T: Have you tested ExpCoT with T≥3? Even synthetic three-step compositions would empirically probe the theorem’s log T promise.

---

> ### Author Response · Authors · 2025-11-20
> **Author Response**
>
> We thank the reviewer for acknowledging that (i) the observed phenomenon is clear and interesting, (ii) the empirical probing is careful, (iii) the theoretical insight is meaningful, and (iv) ExpCoT is simple and effective.
>
> ## Weakness 1: More realistic QA benchmark
>
> We use synthetic tasks precisely because they allow us to ensure operators and their meanings are not seen in pretraining, so that the simple skills must be learned from in-context examples rather than recalled from training data. This fits our purpose of investigating the model’s in-context composition ability, i.e., addressing queries with unknown compositions of unknown skills that are demonstrated only via examples.
> The synthetic tasks also control which basic skills are required and how they are composed. This controlled setting is essential for drawing conclusions about the cause of the failure mode (misalignment of examples and composition steps), rather than entangling it with unrelated sources of error.
>
> ## Weakness 2: exclude the strongest closed models
>
> See Global Response: Other powerful models. In summary, our experimental results on GPT4.1 are consistent with those in our submission. We will perform more experiments and include the new results in the revision.
>
> ## Weakness 1 and Question 1: Scenarios for T > 2
>
> See Global Response: Additional results for compositions with more than 2 simple tasks (T > 2). In summary, the results are in general consistent with those for T=2: (1) more examples from simple tasks may not lead to better performance on composite queries; (2) ExpCoT helps alignment and improves performance.

---

### Official Review · Reviewer_y8gz · 2025-11-01

**Soundness:** 2
**Presentation:** 2
**Contribution:** 1
**Rating:** 2
**Confidence:** 5

**Summary:**

This paper studies the in-context composition ability of large language models (LLMs)—that is, whether models can integrate multiple simple skills demonstrated in the prompt to solve composite tasks requiring multi-step reasoning.
Through controlled experiments on linguistic and logical composition benchmarks (adapted from Xu et al., 2024b) and across 12 open-source LLMs (LLaMA-1/2/3, Mistral, Mixtral, DeepSeek-Distill), the authors find a counterintuitive phenomenon: adding more simple-task examples can hurt performance on composite queries, while additional composite examples help.

The paper attributes this to models’ failure to recognize the compositional structure of tasks—relying on operator matching rather than aligning sub-skills to proper steps. Chain-of-Thought (CoT) prompting does not mitigate this.
Theoretical analysis formalizes conditions under which ignoring compositional alignment increases error, motivating a method called Expanded Chain-of-Thought (ExpCoT), which adds explicit step-wise markers (“Step 1”, “Step 2”, …) to align examples with compositional stages. ExpCoT yields consistent performance improvements across model families.

**Strengths:**

* The paper provides multi-view analysis—performance trends, output correspondences, attention-similarity visualization, and ablation on operator vs. content cues—all supporting a consistent explanation.

* The theoretical and empirical sections are well integrated.

**Weaknesses:**

* Missing and Related Work: The paper omits citation and comparison with Skills-in-Context Prompting: Unlocking Compositionality in Large Language Models, which introduced a closely related framework for enabling compositional reasoning via “skills-in-context” demonstrations. That work similarly examined how combining basic skills in a single prompt affects generalization and proposed structured prompting strategies. The present submission should explicitly position itself relative to SKiC, clarifying differences in (a) task design, (b) prompt structure (ExpCoT vs. SKiC).

* The models tested exclude frontier systems (GPT-4/5, Claude, Gemini), leaving open whether newer models overcome these failures.

* The attention analysis (Fig. 5) is largely qualitative. Quantitative probing (e.g., attention entropy, head attribution) would strengthen causal claims. The CoT baseline could include Decomposed Prompting, Skills-in-Context Prompting for a more complete comparison.

**Questions:**

Could you discuss how ExpCoT compares to Skills-in-Context Prompting in goals, structure, and mechanisms. Is ExpCoT conceptually a structured version of SKiC, or does it introduce new theoretical insights (e.g., compositional alignment proofs)?

Can ExpCoT be tested on more complex or naturalistic tasks (e.g., program synthesis, arithmetic, symbolic reasoning) to demonstrate general utility?

Provide qualitative examples where ExpCoT fails, to better characterize remaining limitations.

---

> ### Author Response · Authors · 2025-11-20
> **Response by Authors: Part 1/2**
>
> We thank the reviewer for recognizing our multi-view and well integrated theoretical and empirical studies. We address the concerns below.
>
> ## Weakness 1 and Question 1: Missing related work [1]
>
> Please see the Global Response: Comparison with related work Skills-in-Context. In summary, [1] is related, while our work has a different research focus and thus a different setting.
>
> Additionally, we would like to compare ExpCoT to SKiC prompting from [1].
> - ExpCoT is more restricted. Besides examples, SKiC provides descriptions about the skills, and also provides step-wise explanation about how to compose the skills to solve the composition task. ExpCoT only provides annotations to the examples about which step uses which skills, so can be regarded as a much restricted version of SKiC. On the other hand, the compositional tasks in [1] can be more complex than ours. This difference stems from the difference in our setting and research focus, as explained in the Global Response.
> - The goal of introducing ExpCoT is also different from that of SKiC. ExpCoT is mainly for analysis and diagnosis of our insights for models’ performance facing unknow compositions of unknown skills demonstrated via in-context examples, while SKiC aims to improve performance by designing prompts explaining the composition to unlock the ability of the model. Note that our main contribution is analysis but not proposing algorithms (see our discussion in the second paragraph of Section 5), and ExpCoT is to confirm our insights from the analysis match the empirical observations.
>
> [1] J. Chen et al. 2024. Skills-in-Context: Unlocking Compositionality in Large Language Models. In Findings of the Association for Computational Linguistics: EMNLP 2024.
>
>
> ## Weakness 2: exclude frontier systems
>
> See Global Response: Other powerful models. In summary, our experimental results on GPT4.1 are consistent with those in our submission. We will perform more experiments and include the new results in the revision.
>
> ## Weakness 3: Quantitative probing
>
> The following table shows the average similarity ± standard deviations. These correspond to the setting of Figure 5 in our submission, i.e., they are for the cosine similarities between the attention maps of some layer of Mistral-7B on a pair of queries. The results show that the similarities between a simple query and a composite query are very high, and are similar to those between two simple queries, or those between two composite queries.
>
> **Table: Similarities of attentions on two queries**
>
> | Layer | 1 | 10 | 15 | 17 | 19 | 25 | 30 | 32 |
> |-------|---|----|----|----|----|----|----|-----|
> | simple-composite | 0.9997 ± 0.0002 | 0.9984 ± 0.0023 | 0.9935 ± 0.0104 | 0.9904 ± 0.0142 | 0.9876 ± 0.0169 | 0.9841 ± 0.0185 | 0.9836 ± 0.0173 | 0.9826 ± 0.0178 |
> | composite-composite | 0.9998 ± 0.0002 | 0.9988 ± 0.0019 | 0.9950 ± 0.0084 | 0.9922 ± 0.0125 | 0.9897 ± 0.0148 | 0.9865 ± 0.0160 | 0.9860 ± 0.0150 | 0.9853 ± 0.0153 |
> | simple-simple | 0.9998 ± 0.0002 | 0.9986 ± 0.0021| 0.9948 ± 0.0089 | 0.9920 ± 0.0125 | 0.9895 ± 0.0169 | 0.9861 ± 0.0169 | 0.9853 ± 0.0161 | 0.9843 ± 0.0169 |
>
> The next table further shows the distributions of the entropy values of the attentions. For a query and a fixed layer, for each attention head we compute the entropy value of its attentions from the query end token to all the tokens in the context, and then compute the histogram of the entropy values of different heads. The rows of the table show the histogram averaged over 100 simple queries or 100 composite queries. The results show that the entropy distributions for the simple queries are similar to those for the composite queries, suggesting that the models does not distinguish between the two types of queries.
>
> **Table: The distribution of the entropy values of the attentions on different types of queries**
> | Histogram Bins | 0-0.5 | 0.5-1 | 1-1.5 | 1.5-2 | 2-2.5 | 2.5-3 | 3-3.5 | 3.5-4 | 4-4.5 | 4.5-5 | 5- |
> |---|---|---|---|---|---|---|---|---|---|---|---|
> | Layer15-Simple | 3.61 | 8.73 | 5.81 | 11.66 | 12.26 | 16.11 | 18.39 | 12.50 | 8.21 | 2.72 | 0.00 |
> | Layer15-Composite | 3.61 | 6.85 | 7.25 | 11.46 | 10.22 | 14.50 | 15.75 | 16.83 | 10.38 | 3.17 | 0.00 |
> | Layer17-Simple | 0.12 | 8.45 | 23.12 | 23.52 | 11.86 | 11.46 | 13.78 | 5.69 | 2.00 | 0.00 | 0.00 |
> | Layer17-Composite | 0.76 | 10.42 | 23.52 | 23.48 | 12.98 | 10.46 | 10.30 | 3.49 | 4.17 | 0.44 | 0.00 |
> | Layer19-Simple | 0.08 | 11.30 | 29.45 | 17.75 | 12.90 | 10.54 | 7.81 | 9.50 | 0.68 | 0.00 | 0.00 |
> | Layer19-Composite | 0.64 | 14.18 | 30.45 | 16.35 | 11.10 | 9.13 | 8.49 | 8.61 | 1.04 | 0.00 | 0.00 |

---

> ### Author Response · Authors · 2025-11-20
> **Author Response: Part 2/2**
>
> ## Question 2: More complex or naturalistic tasks (e.g., program synthesis, arithmetic, symbolic reasoning)
>
> We mainly adopt synthetic tasks from [2], which provide controlled, well-understood benchmarks essential for systematically investigating compositional mechanisms (see discussion in Line 100). Their simplicity enables us to isolate and analyze specific failure modes, such as operator vs. content matching, attention patterns. While we agree that adding original tasks could be valuable for future work, our focus is on understanding the phenomenon, and the controlled clean testbeds fit better for this purpose.
>
> [2] Xu et al. Do large language models have compositional ability? An investigation into limitations and scalability.COLM, 2024.
>
> ## Qustion 3: Failure Cases
>
> We conducted a detailed error analysis to characterize the failure modes of ExpCoT.
>
> We define the following types of outputs:
> - **Correct**: For example, the output is `step1: * Morning Live # → step2: Evening Die # → step3: Die Evening`. Here * denotes opposition and # denotes swap.
> - **Answer_simple1**: Outputs only performs simple task 1. For example, `step1: * Hire Lend # → step2: Borrow Fire # → step3: Borrow Fire`. Here * denotes opposition and # denotes swap.
> - **Answer_simple2**: Outputs only perform simple task 2, e.g., `step1: * occur # → step2: occur # → step3: OCCUR`. Here * denotes past tense and # denotes capitalization.
> - **Special_token**: Outputs "???" at Step 3, e.g., `step1: * Grow Present # → step2: Shrink Past # → step3: ???`. Here * denotes opposition and # denotes swap.
> - **No_Step3**: Fails to generate Step 3 entirely
> - **Step3_noexecute**: Step 3 copies Step 2, e.g., `step1: 79 bake → step2: 79 baked → step3: 79 baked`. Here simple task 1 is plus one to the number and and task 2 is past tense to verb.
> - **Other Faults**: Miscellaneous errors
>
> The following table shows the distributions of the outputs for different number k. The results are averaged over all tasks and all models tested with k_c = 2.
> The key findings are: (1) The most common failure is outputting special tokens (11-13%), suggesting models recognize the need for Step 3 but cannot determine the operation. (2) Simple task confusion is rare (<2%), showing ExpCoT helps recognize compositional nature. The primary remaining challenge is helping models correctly execute aligned steps.
>
> **Table: Distribution of output types across different k values**
>
> | Output Type | k=2 | k=5 | k=8 | k=10 | k=12 | k=15 | k=20 | k=30 |
> |------------|-----|-----|-----|------|------|------|------|------|
> | Correct | 71.12 | 72.81 | 73.74 | 73.42 | 73.31 | 73.76 | 74.40 | 72.21 |
> | Answer_simple1 | 0.12 | 0.07 | 0.05 | 0.05 | 0.10 | 0.16 | 0.14 | 0.20 |
> | Answer_simple2 | 1.20 | 1.32 | 1.12 | 1.49 | 1.32 | 1.71 | 1.43 | 1.90 |
> | Special_token | 13.02 | 12.52 | 11.87 | 12.10 | 11.60 | 11.31 | 11.56 | 11.61 |
> | No_Step3 | 0.04 | 0.02 | 0.02 | 0.03 | 0.00 | 0.00 | 0.01 | 0.06 |
> | Step3_noexecute | 0.15 | 0.07 | 0.08 | 0.07 | 0.03 | 0.07 | 0.09 | 0.11 |
> | Other Faults | 14.34 | 13.18 | 13.11 | 12.85 | 13.65 | 13.00 | 12.37 | 13.91 |
>
>
> The results are similar across different models. To show this, the following table presents the distributions for a specific model Llama2-13b-hf. (Results for other models are also similar and we include only one for brevity.)
>
> **Table: Distribution of output types of Llama2-13b-hf across different k values**
>
> | Output Type | k=2 | k=5 | k=8 | k=10 | k=12 | k=15 | k=20 | k=30 |
> |---|---|---|---|----|----|----|----|-----|
> | Correct | 71.64 | 79.92 | 81.05 | 80.97 | 81.53 | 81.48 | 79.76 | 77.45 |
> | Answer_simple1 | 0.00 | 0.00 | 0.00 | 0.00 | 0.11 | 0.22 | 0.11 | 0.00 |
> | Answer_simple2 | 1.44 | 1.07 | 1.54 | 1.41 | 1.32 | 2.23 | 1.85 | 3.31 |
> | Special_token | 14.83 | 6.01 | 4.58 | 4.67 | 3.53 | 0.77 | 3.35 | 3.47 |
> | No_Step3 | 0.00 | 0.00 | 0.00 | 0.00 | 0.00 | 0.00 | 0.11 | 0.11 |
> | Step3_noexecute | 0.14 | 0.00 | 0.00 | 0.00 | 0.13 | 0.00 | 0.51 | 0.24 |
> | Other Faults | 11.95 | 13.01 | 12.83 | 12.95 | 13.37 | 12.30 | 14.30 | 15.42 |

---

### Official Review · Reviewer_yrht · 2025-11-01

**Soundness:** 3
**Presentation:** 3
**Contribution:** 3
**Rating:** 2
**Confidence:** 4

**Summary:**

This paper investigates the in-context composition ability of LLMs on linguistic and logical tasks. It results reveal that simple task examples can have a surprising negative impact on the performance, because the models generally struggle to recognize and assemble the skills correctly. It further provides theoretical analysis that shows it is crucial to align examples with corresponding steps in the composition.

**Strengths:**

In-depth study of in-context composition of LLMs are important to understanding the behaviors of LLMs in important functionalities like long chain-of-though reasoning or agent tasks.

**Weaknesses:**

1. The paper does not cite and compare with a few important related work such as:

[1] J. Chen et al. 2024. Skills-in-Context: Unlocking Compositionality in Large Language Models. In Findings of the Association for Computational Linguistics: EMNLP 2024.

2. The paper ignores to analyze the in-context composition behavior in the recently emerged long-CoT reasoning LLMs. It would be interesting to analyze if the “thinking” process involves the composition of such elementary skills.

3. The paper could be further improved if in-sights can be added on how the pre-training, mid-training, and post-training (SFT & RL) can improve the LLM’s in-context compositional capabilities.

**Questions:**

None.

---

> ### Author Response · Authors · 2025-11-20
> **Response by Authors**
>
> We thank the reviewer for recognizing that in-depth study of our work and the problem we investigate is important. We address the concerns below.
>
> ## Weakness 1: Comparison with related work [1].
>
> Please see Global Response: Comparison with related work Skills-in-Context. In summary, [1] is related, while our work has a different research focus and thus a different setting.
>
> ## Weakness 2: recently emerged long-CoT reasoning LLMs
>
> We added results by GPT4.1, which are consistent with those in our submission. See the Global Response: Other powerful models.
>
> Our original experiments did include some recent models that are able of long CoT, such as Deepseek-distill-qwen8B and Deepseek-distill-Llama-70b. We would like to note that our work focuses on open-source models for which we can systematically control model sizes, families, and prompting variants, and even inspect the internal attentions. The purpose is to identify the key factors in a transparent setting. On the other hand, we do agree that adding more models can strengthen our work and we will update the draft with the new results.
>
>
> ## Weakness 3: pre-training, mid-training, and post-training (SFT & RL)
>
> While our current work focuses on the test-time in-context composition performance of the model and is not directly on the training, our analysis can provide some hints on the potential roles of pre-training/SFT/RL. Below are our understanding and conjectures, which can be an interesting direction for future work.
>
> - Pretraining primarily equips the model with basic skills and familiarizes it with patterns of text. Our synthetic setting is explicitly constructed so that operators are not seen in pretraining, ensuring that the simple tasks must be learned in-context, isolating the learning of composition from pretraining.
>
> - SFT often introduces structured reasoning formats (e.g., “Step 1, Step 2…,” theorem-style proofs, code comments), which likely encourage the model to internalize the step-wise organization seen in SFT. From our theory, such a step structure directly supports better compositional alignment, if the query follows the same compositional structure as in SFT.
>
> - RL can further reward trajectories that exhibit correct multi-step reasoning, and penalize hallucinated or misaligned steps. Different from SFT, it may not rely on specific step-wise composition formats in the training data. Instead, it may encourage the model to internalize the recognition of composition and the retrieval of correct skills, and thus lead to better performance on compositions unseen in training.
>
> [1] J. Chen et al. 2024. Skills-in-Context: Unlocking Compositionality in Large Language Models. In Findings of the Association for Computational Linguistics: EMNLP 2024.

---

> > ### Comment · Reviewer_yrht · 2025-11-24
> >
> > Thanks for the responses. However, I would prefer to keep my original rating as 2 due to the following reasons.
> >
> > 1. Reading from the rebuttal, it seems this paper is an incremental research based on [1]. In such sense, the introduction and novelty scope of the paper should be significantly revised to meet its actual contribution.
> >
> > 2. The added results (e.g., gpt-4.1) do not provide additional insights on whether/how the compositional skills would affect the long-CoT reasoning or RL-triggered model thinking.

---

> > > ### Author Response · Authors · 2025-11-25
> > > **Further Response**
> > >
> > > We respectively disagree with the comments.
> > >
> > > ## Our work is not based on [1]
> > >
> > > As discussed in detail in the “Global Response: Comparison with related work Skills-in-Context”, we clarify that our work is in a different setting and has different contributions.
> > >
> > > 1. Different settings.
> > > Our work considers unknown compositions of unknown skills demonstrated via examples. [1] considers known compositions of known skills, and thus can provide both examples and detailed explanations about the skills required for the query, and step-by-step descriptions about the composition of the skills to solve the composite query (see Figure 1 in [1]).
> > > The two settings are fundamentally different.
> > >
> > > 2. Different contributions.
> > > Our main contributions are the discovery of the negative impact of more simple task examples, investigation into the reasons (failure to recognize the composition or align the correct skills with the proper steps in the composition), theoretical analysis and an inspired algorithm verifying our insights.
> > > The contributions of [1] are designing a prompt in their setting to unlock the composition ability. Their setting assumes the composition and the skills used are known, and thus the designed prompt can explicitly provide carefully structured information about the compositional structure of the query, and details about which step uses what skills.
> > >
> > > None of our contributions is based on [1]. Our setting considers unknown compositions of unknown skills demonstrated only via examples. Our setting is thus more challenging with novel challenges  (e.g., recognition and misalignment issues). Such issues won’t exist in the setting of [1], and our contributions precisely address these novel challenges and their reasons.
> > >
> > > Given that even the settings are fundamentally different, it is unjustified to claim that our work is based on [1], or incremental research.
> > >
> > > Please provide details about where the reviewer believes our work is based on [1]. We are happy to give further clarifications.
> > >
> > >
> > > ## Our work provided insights into the in-context composition abilities in LLMs with long-CoT reasoning abilities
> > >
> > > First, our work has tested models with long-CoT reasoning abilities, e.g., Deepseek-distill-qwen8B and Deepseek-distill-Llama2-70b. They are investigated throughout our work, and our main contributions include analysis and insights on these models. In short, recognition and misalignment issues prevent the full utilization of the reasoning abilities.
> > >
> > > 1. The models may not recognize the composition. See Section 3.1-3.4.
> > > 2. CoT examples help the recognition, and the models will try to retrieve skills for the steps in the reasoning chain, but the models may misalign the skills with the steps (See section 3.5). Hints on the alignment (e.g., ExpCoT) can significantly help the correct reasoning chain (Section 4.1). In particular, please see Line 344 for an example of why CoT reasoning fails due to misalignment.
> > >
> > > Second, we further added GPT4.1, showing that even powerful recent closed models can have the same issues, regardless the long-CoT reasoning abilities. This provides further support for our insights mentioned above.
> > > We would also like to point out that our current work focuses on the test-time in-context composition performance of the model, regardless of whether the model has RL-triggered modeling thinking or not.
> > >
> > > [1] J. Chen et al. 2024. Skills-in-Context: Unlocking Compositionality in Large Language Models. In Findings of the Association for Computational Linguistics: EMNLP 2024.

---

### Author Response · Authors · 2025-11-20
**Global Response: Part 1/2**

We thank all the reviewers for appreciating the importance of the investigation on in-context composition of LLMs (yrht), the discovery of the counter-intuitive observations (aP4i, eFa6), the systematic and multi-view investigations (y8gz, aP4i), the consistent conclusions with well-integrated empirical and theoretical studies (y8gz, aP4i, eFa6), and a simple fix (aP4i, eFa6).

We address some common concerns below, while discussing others in replies to individual reviewers.

## Additional results for compositions with more than 2 simple tasks (T > 2)

As suggested by the reviews, we added experiments on some compositions of T simple tasks with T=3. The results are in general consistent with those for T=2: (1) more examples from simple tasks may not lead to better performance on composite queries; (2) This is due to misalignment, which can be mitigated by ExpCoT, leading to improved performance. Details are as follows.

We tested 4 compositions: opposition+swap+capitalization, opposition+pastTense+capitalization, pastTense+swap+adding bracelet, pastTense+capitalization+reverse. (See Appendix B.1 in our submission for details of each simple task.)

1. More examples from simple tasks may not help

The table below shows the changes in the accuracy when increasing k, the number of examples from each simple task. We still observe that increasing the number of simple task examples may lead to worse performance. Some models are less affected, but in cases, their performance still drops, e.g., for Llama2-70B, from k=12 to k=15, the accuracy drops from 44.05% to 42.67%.

**Table: Accuracy (in %) on compositions of 3 tasks**

| Model | k=2 | k=5 | k=8 | k=10 | k=12 | k=15 | k=20 |
|---|---|---|---|---|---|---|---|
| Llama-7B | 23.54 | 22.67 | 22.95 | 19.95 | 18.95 | 19.02 | 13.97 |
| Llama-13B | 32.03 | 32.87 | 32.13 | 31.29 | 30.22 | 29.80 | 25.01 |
| Llama-30B | 35.52 | 35.23 | 31.03 | 29.69 | 28.90 | 26.48 | 25.64 |
| Llama-65B | 40.81 | 42.34 | 39.39 | 39.99 | 39.77 | 39.67 | 39.81 |
| Llama2-7B | 29.25 | 30.14 | 30.31 | 31.93 | 32.22 | 33.01 | 28.84 |
| Llama2-13b | 39.02 | 42.50 | 42.34 | 41.76 | 40.37 | 42.64 | 41.31 |
| Llama2-70B | 42.67 | 42.93 | 43.09 | 43.16 | 44.05 | 42.67 | 45.07 |
| Mistral-7B | 42.31 | 44.23 | 44.42 | 44.95 | 43.78 | 45.57 | 45.03 |
| Mistral-8x7B | 43.45 | 43.31 | 43.00 | 42.91 | 43.65 | 42.88 | 44.67 |


2. ExpCoT improves the performance

We also apply ExpCoT for these compositions. The table below shows the performance without ExpCoT v.s. with ExpCoT (using k_c=5 examples from the composition task and k = 15 examples from each simple task).

**Table: Accuracy (in %) on compositions of 3 tasks, without ExpCoT (Vanilla ) v.s. With ExpCoT**

|  | Llama-7B | Llama-13B | Llama-30B | Llama-65B | Llama2-7B | Llama2-13B | Llama2-70B | Mistral-7B | Mistral-8x7B |
|---|---|---|---|---|---|---|---|---|---|
| Vanilla | 19.0 | 29.8 | 26.5 | 39.7 | 33.0 | 42.6 | 42.7 | 45.6 | 42.9 |
| ExpCoT | 44.2 | 38.8 | 54.8 | 61.7 | 50.9 | 55.9 | 64.8 | 57.7 | 67.9 |

Similar to the T=2 cases, ExpCoT consistently improves the accuracy by a large margin, e.g., 25% for Mistral-8*7B. This shows that in the vanilla setting the model indeed suffers from misalignment, while ExpCoT provides hints for the model to better align skills with steps in the composition, and thus improves the performance.



## Other powerful models

We added experiments using GPT4.1 and Gemini2.5, whose results are consistent with our analysis.

The following table shows their results on the task opposition+swap, with k examples from simple task and k_c=5 examples from the composite task (the same setting as in Figure 1 in our submission).
1. We observe that for GPT4.1 without ExpCoT (vanilla setting), initially (from k=2 to 8), increasing k leads to better performance, while later increasing k leads to worse performance. Gemini2.5 also shows a similar pattern. This shows that the model still cannot fully recognize the composition structure and align examples with the correct steps in the composition, but it may partially exploit the examples: at first the benefit of exploiting the examples to infer the skills outweighs the negative impact of misalignment, but later the negative impact dominates.
2. ExpCoT improves the performance significantly. This shows that the hints on aligning the skills and composition steps can mitigate the misalignment issue and thus improve the accuracy.

**Table: Accuracy (in %) of some closed models for different number k of simple task examples**
| | k=2 | k=5 | k=8 | k=10 | k=12 | k=15 | k=20 | k =30 |
|---|---|---|---|---|---|---|---|---|
| GPT4.1 (Vanilla) | 20 | 28 | 34 | 33 | 25 | 25 | 27 | 22 |
| GPT4.1 (ExpCoT) | 65 | 64 | 73 | 69 | 59 | 66 | 69 | 66 |
| Gemini2.5 (Vanilla) | 42 | 40 | 53 | 56 | 53 | 56 | 48 | 57 |
| Gemini2.5 (ExpCoT) | 69 | 70 | 72 | 75 | 70 | 75 | 74 | 69 |

We will continue to perform experiments on more tasks/models and report the results.

---

### Author Response · Authors · 2025-11-20
**Global Response: Part 2/2**

## Comparison with related work Skills-in-Context

Thanks for pointing out the important related work [1]. We will add the citation and the comparison in the revised version. Below, we would like to point out the key difference: our work has a different research focus from the paper. We also note that the paper is also along the general research direction of composition in in-context learning, and their results are consistent and thus supports our analysis.


First, the settings are different:
- Our work focuses on the composition of skills that are *only demonstrated via in-context examples*. The context only consists of in-context examples (without in-context explanations of the skills), where the basic skills are only indicated using operators such as #.
- The paper [1] also studied composing skills in-context, with the focus of unlocking the compositionality ability of the model, by providing *carefully designed Skills-in-Context Prompting* (illustrated in Fig. 1 in the paper). The context in the prompt includes explanations of the basic skills along with examples, and also step-by-step explanations about how to compose them to solve the compositional query.


Second, the different settings stem from different research goals.
- Our work aims to investigate the models’ ability in *a prior unknown composition with a prior unknown skills*. We consider the following scenario: the model can face queries that require compositions of skills, but the required skills and the composition structure are not explicitly specified; only some demonstration examples are available, and the model needs to infer what skills to use and how to compose them using the demonstrations. This leads to the more challenging and constrained setting with only in-context examples but no other contextual information. This facilitates our investigation of the models' ability in retrieving and composing unknown skills using only demonstrations in inference time.
- The paper [1] aims to elicit compositional generalization capabilities of the model. It thus considers prompts that explain and demonstrate the foundational skills and their compositions for accomplishing the composition tasks. Such a prompt structure provides more useful information for the model to achieve composition. As pointed out by the paper, it can allow the model to more actively utilize pre-existing internal skills from pretraining for compositional tasks.


Third, the results from [1] are consistent with the insights from the analysis in our work.
- The prompt provides guidance for the model to learn about steps and skills to use in each step, and thus gets better performance. This is consistent with our experiments and theory, e.g., our theory shows that explicitly aligning skills with steps significantly improves the performance bound; similar hints in ExpCoT on which skills to use in which step can also improve the performance.

[1] J. Chen et al. 2024. Skills-in-Context: Unlocking Compositionality in Large Language Models. In Findings of the Association for Computational Linguistics: EMNLP 2024.

---

### Author Response · Authors · 2025-11-21
**Revision summary**

We have uploaded a revised draft (updates in the main body marked by blue color). The major updates include:

- Added additional results for compositions with more than 2 simple tasks, as suggested by Reviewers aP4i and eFa6. See Line 161 and Appendix B.9.
- Added additional results from GPT4.1 and Gemini2.5, as suggested by Reviewers yrht, y8gz, and aP4i. See Line 162 and Appendix B.10.
- Added citations and discussion for missing related work, as suggested by Reviewers yrht, y8gz, and eFa6. See Line 109 and Appendix A Line 1039. We also updated the title/abstract/introduction slightly to reflect the difference from existing works.
- Added more quantitative probing for attentions, as suggested by Reviewer y8gz. See Line 323 and Appendix B.5.
- Added failure case analysis for ExpCoT, as suggested by Reviewer y8gz. See Line 467 and Appendix B.8.1.

We thank the reviewers for the suggestions. Please let us know if there are more suggestions and we are happy to engage on further discussions.

---

### Author Response · Authors · 2025-12-03
**Author Final Remarks**

Dear AC and Reviewers,

We thank you for your constructive feedback, which has strengthened our work. Our work investigated the composition abilities of LLMs to perform novel composite tasks that combine unknown simple task skills demonstrated only in in-context examples. We discovered that more examples from simple tasks can decrease the performance and provided systematic empirical and theoretical analyses. We also made partial progress towards algorithm improvement by providing a simple and effective method for our synthetic tasks.

## Key Strengths Highlighted:
- **The importance of the investigation on in-context composition of LLMs (yrht)**
- **The discovery of the counter-intuitive observations (aP4i, eFa6)**
- **The systematic and multi-view investigations (y8gz, aP4i)**
- **The consistent conclusions with well-integrated empirical and theoretical studies (y8gz, aP4i, eFa6)**
- **A simple fix (aP4i, eFa6)**

## Concerns Addressed:
- **Compositions with more than 2 simple tasks (aP4i, eFa6)**: We added experiments on compositions of 3 simple tasks. The results are consistent with those in the original submission: (1) more examples from simple tasks may not lead to better performance on composite queries; (2) This is due to misalignment, and can be mitigated by ExpCoT, leading to improved performance.
- **Other powerful models (yrht, y8gz, aP4i)**: We added experiments using closed models GPT4.1 and Gemini2.5, providing more supports for our analysis.
- **Comparison with missing related work (yrht, y8gz, eFa6)**: We added discussions with the missing related work pointed out. We clarified that our setting is fundamentally different, which stems from the different research focus.

Other questions/concerns:
- **Rev. y8gz**: asked for more quantitative probing on the attentions and failure cases of ExpCoT. We added detailed statistics about the similarities and entropy, and provided both qualitative examples and quantitative analysis of the failure cases.
- **Rev. eFa6**: asked for performance on the simple tasks. We clarified that this has been reported in previous work.

We hope this summary clarifies the reviewers’ positions and highlights our efforts to engage constructively with their feedback. Thank you once again.

Sincerely,

The Authors

---

### Meta-Review · Area_Chair_P7f6 · 2026-01-07

**Summary:**

This paper offers a careful study of in-context skill composition in LLMs on controlled linguistic and logical benchmarks, highlighting a counterintuitive observation where adding simple exemplars can degrade composite-task performance. The empirical analyses are generally good, and the proposed alignment strategy is effective. However, some reviewers have serious concerns that the work misses key related baselines, does not test newer long-CoT/frontier models, and is limited mostly to synthetic two-step compositions without clear evidence of robustness for longer or more realistic reasoning tasks.

**Reviewer Concerns:**

Reviewers raised concerns about the absence of key baselines, the lack of evaluation on newer long-CoT or frontier models, and the study’s reliance primarily on synthetic two-step composition tasks.

**Reviewer Scores:**

Reviewers generally keep their scores after rebuttal.

---

### Decision · Program_Chairs · 2026-01-26

Reject